# Controlling Language and Diffusion Models by Transporting Activations

**Pau Rodríguez**[*] **Arno Blaas** **Michal Klein** **Luca Zappella** **Nicholas Apostoloff**
**Marco Cuturi** **Xavier Suau**[*]
{pau.rodriguez,ablaas,michal_klein,lzappella,napostoloff,
m_cuturi,xsuaucuadros}@apple.com
Apple

## Abstract

The increasing capabilities of large generative models and their ever more widespread deployment have raised concerns about their reliability, safety, and potential misuse. To address these issues, recent works have proposed to control model generation by steering model activations in order to effectively induce or prevent the emergence of concepts or behaviors in the generated output. In this paper we introduce Activation Transport (AcT), a general framework to steer activations guided by optimal transport theory that generalizes many previous activation-steering works. AcT is modality-agnostic and provides fine-grained control over the model behavior with negligible computational overhead, while minimally impacting model abilities. We experimentally show the effectiveness and versatility of our approach by addressing key challenges in large language models (LLMs) and text-to-image diffusion models (T2Is). For LLMs, we show that AcT can effectively mitigate toxicity, induce arbitrary concepts, and increase their truthfulness. In T2Is, we show how AcT enables fine-grained style control and concept negation.

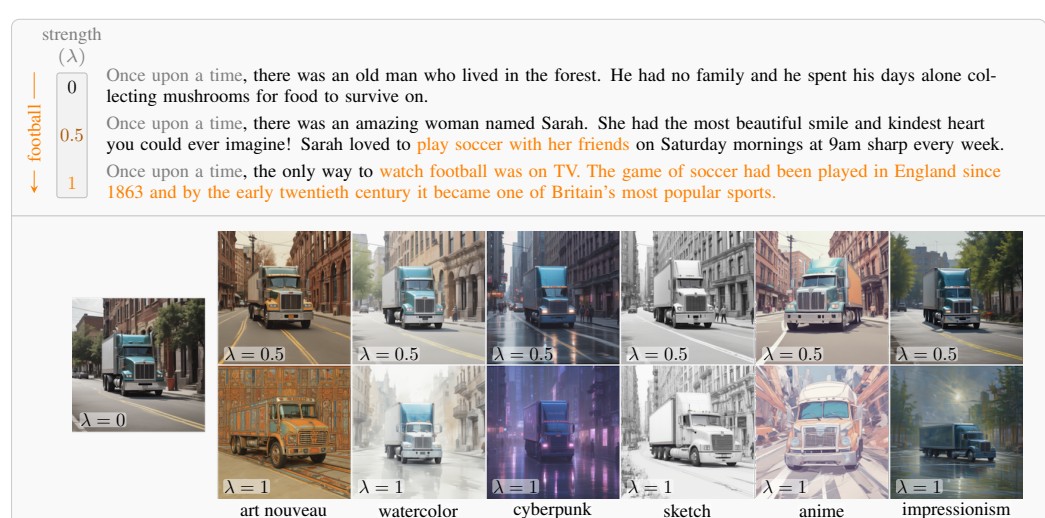

Figure 1: **Linear-AcT unlocks interpretable controllability for both LLMs and Diffusion**, offering explicit control over the strength of conditioning, via a parameter $\lambda$ between 0 (no transport) and 1 (full transport).

---

[*]Equal contribution.

# 1 INTRODUCTION

Pre-trained Generative Models (GMs) typically undergo an additional fine-tuning phase to better align them to a desired behavior. For example, Large Language Models (LLMs) are aligned via instruction fine-tuning (Wei et al.) or RLHF (Ouyang et al., 2022). Although less extensively, these strategies have also been applied to Text-to-Image (T2I) models (Wallace et al., 2024; Yang et al., 2024). However, as the number of parameters grows, alignment approaches can become challenging from a computational and memory perspective (Houlsby et al., 2019). In addition, these strategies modify the model's internal mechanisms, realigning its parameters by leveraging new data, which can have the undesired side effect of impacting the utility of the model on other metrics (Kotha et al., 2024; Luo et al., 2023), such as 0-shot evaluation or question-answering.

The increasing cost of fine-tuning has motivated research in inference-time interventions on pre-trained models that offer a better understanding of features (Geiger et al., 2024) or to control specific behaviors (Suau et al., 2022; Rimsky et al., 2023; Zou et al., 2023; Li et al., 2024). Since these modifications are typically sparse and/or low-dimensional, they can be estimated using a few hundreds of sentences (Suau et al., 2024; Turner et al., 2023). For example, Rimsky et al. (2023); Li et al. (2024) shift activations by a constant vector estimated with sets of desired and undesired data (*e.g.,* non-toxic and toxic); or Suau et al. (2024) mitigate toxicity by dampening the activations of expert neurons. While effective, existing methods do not preserve the activation distribution observed by the model during training. Considering how brittle GMs can be (Huu-Tien et al., 2024; Sclar et al., 2024), a constant shift can move activations out-of-distribution (OOD), which can lead to unwanted behaviors, and hinder both the conditioning and the general model performance.

We propose Activation Transport (AcT), a framework to steer activations according to the optimal transport (OT) map between two different (source and target) activation distributions, *e.g.,* toxic to non-toxic language, or between two different styles in T2I generation. AcT applies a set of univariate maps on activations while preserving their target distributions, achieving better controllability and robustness to the choice of model and layers intervened upon. Our main contributions are:

- A unifying interpretation of existing activation steering methods under the umbrella of OT, showing that most existing methods are equivalent to a mean transport map (Section 3.3).
- Linear-AcT, an inference-time intervention[1] based on OT that preserves internal activation distributions (Section 3.1). The degree of intervention can be controlled by a strength parameter $\lambda$ between 0 (no transport) and 1 (full transport), as shown in Figure 1. We also introduce the *transport support* to prevent inducing OOD activations.
- We show that, without any hyperparameter tuning, Linear-AcT matches or outperforms existing inference-time interventions when aiming to control LLMs for the tasks of toxicity mitigation, concept induction, and increasing truthfulness.
- We find that off-the-shelf Linear-AcT is also effective at controlling T2I diffusion models for the tasks of fine-grained style control and concept negation. Additionally, we adapt (Li et al., 2024) (ITI) for T2I. To the best of our knowledge, this is the first work to apply an inference-time intervention method that is simultaneously effective on both LLMs and Diffusion Models.

# 2 RELATED WORK

The growing capabilities and prevalence of GMs (Brown et al., 2020; Rombach et al., 2022), along with the rising costs of fine-tuning and alignment, have driven research into controllability of GMs.

**Controlling LLMs.** ACTADD (Turner et al., 2023) uses a contrast prompt (one positive and one negative example) to construct a shift vector. CAA (Rimsky et al., 2023) builds on ACTADD by calculating the difference vectors for steering based on a dataset of contrast pairs (rather than a single pair), adding the mean difference during inference time for steering. ITI-C (Li et al., 2024) estimates the shift vector orthogonal to the hyperplane learnt by a binary linear classifier on two sets of sentences, showing an increase of truthfulness on the TruthfulQA benchmark (Lin et al., 2021). The same work proposes MassMean (ITI-M), with an additive vector computed as the difference in means for both sets of sentences. With a different approach, AURA by Suau et al. (2024) dampens activations proportionally to each neuron's ability to classify toxic and non-toxic sentences,

---

[1] Code available at `https://github.com/apple/ml-act`

effectively mitigating toxicity. REPE by Zou et al. (2023) proposes to compute steering vectors at inference time based on prompt pairs. Wu et al. (2024) considers activations relationships using a low-rank projection to exchange information with a counterfactual representation and Geiger et al. (2024) consider rotations of subsets of features. Orthogonal to the works of activation steering, Dekoninck et al. (2023) have proposed a language model arithmetic that can combine the outputs of multiple models in a principled way to simultenously control multiple concepts, however requiring several (costly) inference passes on the LLM.

**Controlling T2I** Few works tackle aligment of T2I models. Wallace et al. (2024) align diffusion models with reinforcement learning (RL) on human comparison data. Yang et al. (2024) remove the need of a reward model to reduce computational overhead of RL. Other works focus on fine-tuning to maximize a reward function (Clark et al., 2023) or consistency to reference images (Lee et al., 2024). The literature on T2I diffusion model controllability is more extensive and it commonly consists in training structure adapters (Mou et al., 2024; Jiang et al., 2024), style adapters (Stracke et al., 2024; Ye et al., 2023; Zhao et al., 2024), or low-rank adapters (LoRAs) (Ruiz et al., 2023; YEH et al., 2024; Gandikota et al., 2023; Stracke et al., 2024). Closer to our work are inference-time interventions, which do not require backpropation through the model to train the conditioning mechanisms. Diffusion steering methods are a family of inference-time interventions, which directly modify the diffusion algorithm at test time for fine-grained control with additional prompts (Nair et al., 2023; Brack et al., 2022). To the best of our knowledge, our work is the first to explore inference-time interventions that are not specific to diffusion models and transfer across modalities.

## 3 TRANSPORTING NEURON ACTIVATIONS

We represent the activations of a GM given an input sentence $x \in \mathcal{S}$ as a tensor $\mathbb{R}^{M \times L \times K}$, where $M$ is the number of activations per layer (assumed constant w.l.o.g. for simplicity), $L$ the number of layers, and $K$ the number of tokens decoded. We reduce each of the $K$ values to only one using an arbitrary pooling operator $\phi$. From now on we write $\mathbf{Z} : \mathcal{S} \to \mathbb{R}^{M \times L}$ for the map that turns a sentence into a matrix of activations statistics, noting that $\mathbf{Z}$ incorporates $\phi$-pooling.

We consider two probability distributions on sentences $p$ and $q$. We view these sentences through the lens of their aggregated activation matrices, *i.e.,* we will examine probability distributions $\mu := \mathbf{Z}\sharp p$ and $\nu := \mathbf{Z}\sharp q$, where we have used the pushforward operator $\sharp$. In practice, we have access to samples $x^1, \ldots, x^n \sim p$ and $y^1, \ldots, y^n \sim q$. For instance, in the case of toxicity mitigation, $p$ covers *toxic* sentences and $q$ *non-toxic* ones. Input sentences $x^i$ and $y^i$ go through the model to yield activation matrices $a^i := \mathbf{Z}(x^i)$ and $b^i = \mathbf{Z}(y^i)$, each seen as i.i.d. samples from $\mu$ and $\nu$ respectively, resulting in $n + n$ observations of $M \times L$ matrices. In that context, our goal is to learn a transport map $T : \mathbb{R}^{M \times L} \to \mathbb{R}^{M \times L}$ from $(a^i, b^i)$ that approximately pushes $\mu$ to $\nu$, *i.e.,* $T\sharp\mu \approx \nu$.

### 3.1 LOW BUDGET ESTIMATORS FOR TRANSPORT MAPS

Since a modern GM can have millions of activations, an ideal transport estimator for $T$ must be easy to learn, cheap to store in memory, and blazing fast to evaluate to avoid overheads at inference time. Additionally, because the estimation of OT maps is known to be plagued by the curse of dimensionality (Chewi et al., 2024, Chap. 2), notable care must be taken to have map estimates that generalize reasonably well. These issues are all compounded by the fact that our final method, as presented in §3.2 builds on a composition of such OT maps (i.e. maps for a layer are estimated on samples that are themselves obtained by using maps for a previous layer). For all these fundamental reasons, we work our way from very simple map estimators, and follow Suau et al. (2024) to focus on maps that factorize *independently* along each dimension (each activation). $T$ is therefore described as a collection of $ML$ independent univariate maps, where each map indexed by $m, l$ should ideally map the marginal distribution of $\mu$ in that coordinate to that of $\nu$. Recall that:

**Proposition 3.1 (Univariate Transport Maps)** *(Santambrogio, 2015, Chap.2) Let $\rho, \tau \in \mathcal{P}(\mathbb{R})$ be two univariate distributions. For any submodular cost $c : \mathbb{R} \times \mathbb{R} \to \mathbb{R}$ (i.e., such that $\partial c / \partial x \partial y < 0$), the optimal transport map $T$ that can transport $\rho$ to $\tau$ is $T^\star = Q_\tau \circ F_\rho$, where $Q_\tau$ and $F_\rho$ are respectively the quantile function of $\tau$ and the cumulant density function (CDF) of $\rho$.*

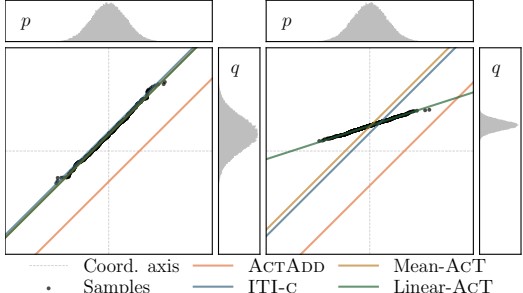
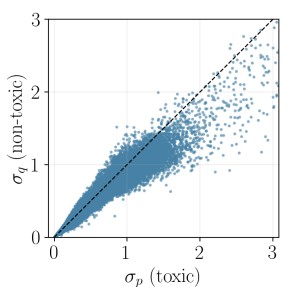
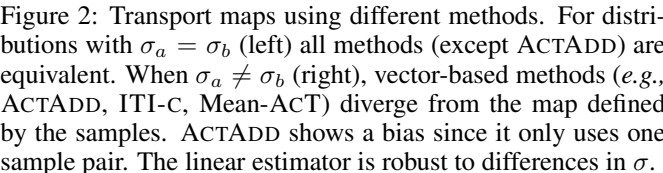

Figure 2: Transport maps using different methods. For distributions with $\sigma_a = \sigma_b$ (left) all methods (except ACTADD) are equivalent. When $\sigma_a \neq \sigma_b$ (right), vector-based methods (*e.g.,* ACTADD, ITI-C, Mean-ACT) diverge from the map defined by the samples. ACTADD shows a bias since it only uses one sample pair. The linear estimator is robust to differences in $\sigma$.

Figure 3: Actual $\sigma_a, \sigma_b$ for toxic and non-toxic sentences on Gemma2-2B, showing that $\sigma_a \neq \sigma_b$ in real scenarios.

Estimating and storing all $ML$ transport maps would therefore require dealing with as many quantile and CDF functions. Unfortunately, parameterizing each of these could quickly become intractable, which is why we scale down ambitions to simplify further our working hypothesis to only consider *affine* transport maps. Each of the $ML$ activations we consider results in two families of reals: source $(a_{m\ell}^1, \ldots, a_{m\ell}^n)$ and targets $(b_{m\ell}^1, \ldots, b_{m\ell}^n)$. Simpifying notations, we drop mentions to $m$ and $\ell$ to focus on values $A := (a^1, \ldots, a^n)$ and $B := (b^1, \ldots, b^n)$ each in $\mathbb{R}^n$. We propose to consider the simple proxy task of finding *affine* maps that push $A$ to $B$ efficiently. We present such an affine map, denoted Linear-ACT, in Definition 3.1. Despite its simplicity, we show in Section 3.3 that many state-of-the-art methods boil down to even simpler approximations and heuristics.

**Definition 3.1 (Linear-ACT)** *Given samples $A = (a^1, \ldots, a^n)$ and $B = (b^1, \ldots, b^n)$ and a cost function $c : \mathbb{R} \times \mathbb{R} \to \mathbb{R}$, the Linear-ACT map trained with these samples is defined as*

$$T(a; A, B) := \omega a + \beta,$$

*where $\omega, \beta$ are the minimizers of $\min_{\omega, \beta} \sum_i c\big(b^{(i)}, \omega a^{(i)} + \beta\big)$, and can be recovered in closed form when $c(a, b) := (a - b)^2$, as*

$$\omega = \frac{\sum_i \tilde{a}^{(i)} \tilde{b}^{(i)}}{\sum_i (\tilde{a}^{(i)})^2}, \quad \beta = m_b - \omega m_a,$$

*where $m_a = \frac{1}{n} \sum_i a^i, m_b = \frac{1}{n} \sum_i b^i$ are mean values, and superscripted values with $^{(i)}$ refer to sorted values in increasing order, $(a^{(1)} \leq \cdots \leq a^{(n)})$ and $(b^{(1)} \leq \cdots \leq b^{(n)})$. Additionally, $\tilde{a}^{(i)} = a^{(i)} - m_a, \tilde{b}^{(i)} = b^{(i)} - m_b$ are sorted and recentered observations.*

An important feature of Linear-ACT is that it can be composed with linear layers in the GM, resulting in no computational overhead at inference time (see Appendix A for details). Note that the expression in Linear-ACT should *not* be confused with the closed-form known when transporting a Gaussian density with parameter $(m_a, \sigma_a)$ towards a second $(m_b, \sigma_b)$, which is known (Peyré & Cuturi, 2019, Remark 2.31) to be $T(a) = \frac{\sigma_b}{\sigma_a} a + (m_b - \frac{\sigma_b}{\sigma_a} m_a)$. Note that if one makes the additional assumption that $\sigma_a = \sigma_b$, then the affine Gaussian map becomes a mean shift or translation, with $T(a) = a + m_b - m_a$. We call this very simple baseline Mean-ACT and show in Section 3.3 that several methods in the literature indeed propose versions of a mean shift strategy.

Figure 2 showcases the effect of different maps on toy data (iid, Gaussian). Note that methods based on mean-shift (ACTADD, ITI-C, Mean-ACT) can strongly over or undershoot, mapping samples out-of-distribution. Linear-ACT shows a good trade-off between in distribution mapping and low computational budget. We note that activations in current GMs show mostly unimodal distributions, but have different standard deviations for different behaviors as shown in Figure 3, making the linear choice a suitable one. Note that multimodal distributions would result in non-linear transport maps, which are beyond the scope of this work.

**Transport Support** The map in Definition 3.1 is estimated using $n$ pairs of samples. In practice, $n$ is in the order of hundreds, which results in a rough approximation of the true transport from $\mu$ to $\nu$. It is fair to assume that the transport error will be higher for input samples in the tail of $\mu$, given the scarcity of samples in that range. Because transporting OOD samples may lead to unexpected behavior, and to be on the conservative side, we only transport new samples that are within the osberved support $\mathcal{Q}_o = [\min A, \max A]$. Using the support is important when $\mu$ is *narrower* than $\nu$ (typically in a mitigation setup). Unless stated otherwise, we use $\mathcal{Q}_o$ for concept mitigation and $\mathcal{Q}_\infty = (-\infty, \infty)$ for induction. Appendix E shows an empirical validation of this choice.

## 3.2 Sequential Iterative Maps

While it might be possible to follow the template approach outlined in Section 3.1 to apply univariate maps to each of the $ML$ activations, this ignores the causal relationship across activations, where activations produced by a layer are processed by the next one, *i.e.,* $\boldsymbol{a}_{m,\ell+1} = f_\ell(\boldsymbol{a}_{m,\ell})$. Any intervention at the level of a layer must therefore be factored in accordingly before creating the intervention at the next one. To account for such causality, we estimate the transport maps for each layer incrementally: we first estimate the transport for the first layer (in the model graph), then we run inference again by applying the first layer map in order to estimate the map for the second layer, and so on until all maps are estimated. A similar approach is adopted in Zou et al. (2023), and detailed with our tools in Definition 3.2. In Appendix C we show that causal estimation achieves more effective conditioning than a simultaneous estimation. In this work, we use causal estimation for Mean-AcT and Linear-AcT.

**Definition 3.2 (Affine Causal Transport Map)** *For* $m \leq M$ *and* $\ell \leq L$, *let* $A_m := (\boldsymbol{a}_{m,1}^1, \cdots, \boldsymbol{a}_{m,1}^n)$ *and* $B_m := (\boldsymbol{b}_{m,1}^1, \cdots, \boldsymbol{b}_{m,1}^n)$ *denote* $n$ *families of* $M$ *activations for the first layer. Starting with* $\ell = 1$, *and setting*

$$C_{m,1} := A_{m,1}, D_{m,1} := B_{m,1},$$

*compute and store the* $2M$ $(\omega_m, \beta_m)$ *parameters of all* $M$ *transport maps associated with these activations using Definition 3.1:*

$$\forall m \leq M, \forall \ell \leq L, \quad T_{m,\ell} := T(\,\cdot\,; C_{m,\ell}, D_{m,\ell}) : \mathbb{R} \to \mathbb{R},$$

*where observations* $C$ *and* $D$ *are refreshed recursively for each of their entries* $m \leq M$, *as* $\ell$ *is incremented,*

$$C_{\cdot,\ell+1} := f_\ell([T_{m,\ell}(C_{m,\ell})]_m),$$
$$D_{\cdot,\ell+1} := f_\ell([T_{m,\ell}(D_{m,\ell})]_m).$$

*At inference time, given a sentence* $\boldsymbol{x}$, *we run the recursion starting from the first activation vector* $\boldsymbol{a} = (\boldsymbol{a}_{m,1})_m$, *looping for* $1 \leq \ell \leq L$ *as* $\boldsymbol{a} \leftarrow f_\ell([T_{m,\ell}(\boldsymbol{a}_m)]_m$.

**Interpolation Between Measures Using Transport** One can easily extend a transport map from measure $\mu$ to $\nu$ to one that is able to output an interpolating measure. The idea, outlined by McCann (1997), consists in defining the following $\lambda$-parameterized map from any OT map $T$,

$$T(a, \lambda) = (1 - \lambda)a + \lambda T(a), \tag{1}$$

where $\lambda \in [0, 1]$ and $\lambda = 1$ recovers the full transport. Conditioning GMs through OT allows the user to precisely control the presence of a concept with a continuous and interpretable *knob* ($\lambda$) during generation, not requiring expensive parameter search (Li et al., 2024) or being limited by fixed, uncontrollable conditioning (Suau et al., 2024). In applications such as diffusion, where the utility of the model is harder to assess, our interpretable strength is of key importance, as shown in Section 5. Note that methods like AcTADD, CAA or ITI-C also have a conditioning strength parameter. However, this parameter is applied as a multiplier of a conditioning bias as $T(a, \lambda) = a + \lambda\beta$ (see Section 3.3), thus making $\lambda$ unbounded, harder to interpret and not robust with respect to different models, layers, and tasks.

## 3.3 Generalization of Prior Inference-Time Interventions Work

In this section, we show how many earlier works can be interpreted as special cases of Linear-AcT. Table 1 summarizes the intervention proposed by several recent methods, where we show that all

Table 1: Comparison of different inference-time interventions in the literature. All methods listed can be expressed as a specific form of a linear map. With AcT, the conditioning strength $\lambda$ interpolated between the activation $a$ and its transformed version (following Equation (1)), while existing methods use $\lambda$ as a bias multiplier, thus becoming less interpretable and less robust to model/layer changes. As a result, many methods require a grid-search to find the best layer to intervene upon.

| Method | Transport | Parameters | Support | $\phi$ |
|---|---|---|---|---|
| Det$_{\text{zero}}$ (Suau et al., 2022) | $\omega a + \beta$ | $\omega = 0, \; \beta = m_b$ | Any layer, $a \mid \text{AP}(A,B) > \varepsilon$ | max |
| ACTADD (Turner et al., 2023) | $\omega a + \lambda \beta$ | $\omega = 1, \; \beta = a^+ - a^-$ | Layer search | last |
| CAA (Rimsky et al., 2023) | $\omega a + \lambda \beta$ | $\omega = 1, \; \beta = m_b - m_a$ | Layer search | last |
| RePE (Zou et al., 2023) | $\omega a + \lambda \beta$ | $\omega = 1, \; \beta = a^+(\boldsymbol{x}) - a^-(\boldsymbol{x})$ | Layer search | last |
| AURA (Suau et al., 2024) | $\omega a + \beta$ | $\omega = 1 - Gini(A,B), \; \beta = 0$ | Any layer, $a \mid \text{AUROC}(A,B) > 0.5$ | max |
| EAST (Rahn et al., 2024) | $\omega a + \lambda \beta$ | $\omega = 1, \; \beta \approx m_b$ | Layer search | last |
| ITI-M (Li et al., 2024) | $\omega a + \lambda \beta$ | $\omega = 1, \; \beta = m_b - m_a$ | Attention head search | last |
| ITI-C (Li et al., 2024) | $\omega a + \lambda \beta$ | $\omega = 1, \; \beta = f_{CLS}(A,B)$ | Attention head search | last |
| Mean-AcT, Section 3.1 | $(1-\lambda)a + \lambda(\omega a + \beta)$ | $\omega = 1, \; \beta = m_b - m_a$ | Any layer, $a \in \mathcal{Q}_o$ or $\mathcal{Q}_\infty$ | mean |
| Linear-AcT, Definition 3.1 | $(1-\lambda)a + \lambda(\omega a + \beta)$ | $\omega, \beta = \arg\min_{\omega,\beta} \sum_i (b^{(i)} - (\omega a^{(i)} + \beta))^2$ | Any layer, $a \in \mathcal{Q}_o$ or $\mathcal{Q}_\infty$ | mean |

methods propose a form of linear transport, and all of them (aside from Suau et al. (2022)) add a bias to the activations. The way this bias is pre-computed is what differentiates each method. Note that the parameter $\lambda$ typically multiplies the bias, thus becoming unbounded and non-interpretable.

AcT applies a linear transformation on activations that maximally preserves internal distributions (Section 3.1, and distribution plots in Appendix F). Moreover, AcT interpolates between the current and transformed activations, making $\lambda$ bounded between $[0,1]$ and interpretable. An additional aspect is that other methods propose various heuristics to choose the support, while AcT uses all activations or the observed input range ($\mathcal{Q}_o$). Note that CAA, ITI-M and Mean-AcT use a difference in means. We subsume this family of methods reporting results for Mean-AcT, which has the additional advantage of an interpretable $\lambda$. An additional difference is that many methods use the last token only (in pseudocode, $\phi(\boldsymbol{z}) = \boldsymbol{z}[\ldots, -1]$). Det$_{\text{zero}}$ and AURA use max-pooling ($\phi(\boldsymbol{z}) = \boldsymbol{z}.\text{max}(-1)$) while AcT uses an average across tokens ($\phi(\boldsymbol{z}) = \boldsymbol{z}.\text{mean}(-1)$), which we have found to be more robust (see Appendix D).

## 4 EXPERIMENTS ON LLMS

We empirically verify the performance of AcT on pre-trained LLMs on toxicity mitigation (Section 4.1), general concept induction (Section 4.2), and truthfulness induction in particular (Section 4.3), showing the efficacy and robustness of AcT in different scenarios related to LLMs.

### 4.1 TOXICITY MITIGATION IN LLMS

It is known that LLMs are prone to generate toxic language (Wen et al., 2023), especially when prompts are designed to elicit toxic behavior. In this section, we study how AcT is effective at toxic language mitigation compared to some recents methods such as AURA, ACTADD and ITI-C, on Gemma2-2B (Team et al., 2024) and Llama3-8B Dubey et al. (2024). To do so, we prompt each LLM with 1000 randomly chosen prompts from RealToxicityPrompts (RTP) (Gehman et al., 2020), known to induce toxic language generation. Then, we collect the generated continuation to each prompt and we evaluate toxicity with a ROBERTA-based classifier[2], as in Suau et al. (2024). In addition, we also measure toxicity in a 0-shot manner by querying Llama3-8B-instruct as LLM-as-a-judge (Zheng et al., 2023) (more details on Appendix H). As a measure of general LLM utility we report in Table 2: (i) perplexity (PPL) on a fixed set of 20k Wikipedia sentences measured with the intervened model, (ii) PPL of the generated sentences measured with Mistral-7B (Jiang et al., 2023) and (iii) MMLU (Hendrycks et al., 2021) 5-shot accuracy using the intervened model. Besides, we report generation diversity results in Appendix G.

**Linear-AcT reduces toxicity up to $7.5\times$ and is robust to $\lambda$, layer, and model choice** We observe that Linear-AcT achieves up to $7.5\times$ reduction in toxicity on Gemma2-2B and $4.3\times$ on Llama3-8B, with minimal impact on PPL and MMLU. Most importantly, AcT obtains the best results at $\lambda = 1$, which is in line with our OT formulation, since $\lambda = 1$ means full transport. Linear-AcT and Mean-AcT obtain similar toxicity mitigation results. ITI-C achieves $5.6\times$ and

---

[2] https://huggingface.co/s-nlp/roberta_toxicity_classifier

Table 2: Toxicity mitigation for Gemma2-2B and Llama3-8B, results over 5 runs. We intervene upon different layer types (layer column) and show the best layer per method. ITI-C, ACTADD and ACT have a *strength* parameter $\lambda$ which we sweep. For each method, we report results for the $\lambda$ that attained the best CLS toxicity that incurs less than $+1$ increase in PPL Wikipedia. ACT methods and provide best results for $\lambda = 1$, achieving up to $7.5\times$ (Gemma2-2B) and $4.3\times$ (Llama3-8B) CLS toxicity mitigation with Linear-ACT. ITI-C is very sensitive to $\lambda$ as well as layer choice (see full results in Appendix J), and AURA reaches up to $3.1\times$ reduction.

|  |  | Layer | Best $\lambda$ | CLS Tox. (%) ↓ | 0-shot Tox. (%) ↓ | PPL Wikipedia ↓ | PPL Mistral-7B ↓ | MMLU ↑ |
|---|---|---|---|---|---|---|---|---|
| **Gemma2-2B** | Original | - | - | $4.17 \pm 0.32$ | $13.42 \pm 1.08$ | 13.98 | 6.68 | 53.1 |
| | ACTADD | MLP | 0.5 | $3.96 \pm 0.24$ (1.1×) | $13.43 \pm 1.42$ | 14.69 (+0.72) | 6.67 (+0.05) | 53.0 (-0.1) |
| | AURA | MLP | - | $2.12 \pm 0.27$ (2.0×) | $9.04 \pm 0.66$ | 14.18 (+0.21) | 7.04 (+0.36) | 53.0 (-0.1) |
| | ITI-C | Attention | 8.0 | $0.74 \pm 0.18$ (5.6×) | $5.36 \pm 0.91$ | 14.90 (+0.92) | 7.44 (+0.76) | 52.6 (-0.5) |
| | Mean-ACT | Post-LN | 1.0 | $\mathbf{0.54 \pm 0.44}$ (7.7×) | $\mathbf{4.10 \pm 0.41}$ | 14.21 (+0.23) | 7.59 (+0.90) | 51.6 (-1.5) |
| | Linear-ACT | Post-LN | 1.0 | $\underline{0.56 \pm 0.21}$ (7.5×) | $\underline{4.14 \pm 0.55}$ | 14.79 (+0.81) | 7.99 (+1.31) | 51.3 (-1.8) |
| **Llama3-8B** | Original | - | - | 5.80 | 15.00 | 9.06 | 5.68 | 65.3 |
| | ACTADD | Attention | 0.3 | $5.57 \pm 0.45$ (1.0×) | $15.73 \pm 0.21$ | 9.71 (+0.65) | 5.85 (+0.16) | 65.5 (+0.2) |
| | AURA | MLP | - | $1.90 \pm 0.61$ (3.1×) | $8.12 \pm 0.85$ | 9.52 (+0.45) | 6.05 (+0.37) | 65.5 (+0.2) |
| | ITI-C | Attention | 3.0 | $1.60 \pm 0.22$ (3.6×) | $\underline{6.53 \pm 0.66}$ | 9.48 (+0.42) | 6.17 (+0.49) | 64.7 (-0.6) |
| | Mean-ACT | Attention | 1.0 | $\underline{1.38 \pm 0.17}$ (4.2×) | $\mathbf{5.60 \pm 0.34}$ | 9.56 (+0.49) | 6.36 (+0.68) | 64.7 (-0.7) |
| | Linear-ACT | Attention | 1.0 | $\mathbf{1.35 \pm 0.39}$ (4.3×) | $6.68 \pm 0.81$ | 9.56 (+0.49) | 6.28 (+0.60) | 64.5 (-0.8) |

$3.6\times$ toxicity reduction on Gemma2-2B and Llama3-8B respectively. In line with the ITI-C paper findings, ITI-C performs well on attention, but is very sensitive to models and layers, as well as to the choice of $\lambda$ (see a layer diagram in Appendix B and full tables and plots in Appendix J). AURA achieves $2.0\times$ and $3.1\times$ toxicity reduction per model and ACTADD induces the mildest mitigation.

## 4.2 Inducing Concepts in LLMs with ACT

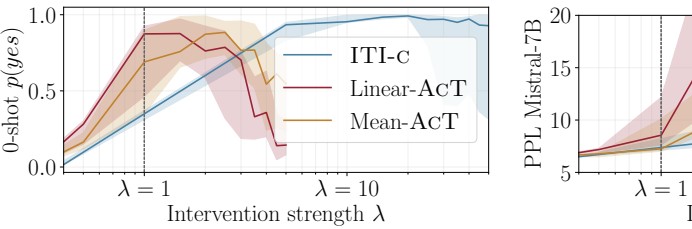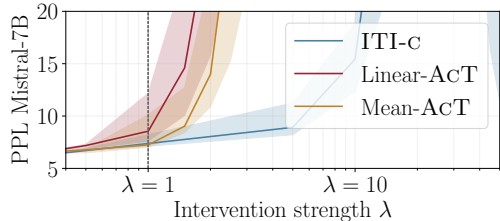

Figure 4: Concept induction using ACT (post-LN layers) and ITI-C (attention layers) on Gemma2-2B. We aggregate results over 7 WordNet concepts, generating 500 sentences at different intervention strength levels. We report concept presence with LLM-as-a-judge ($p(yes)$), and the PPL of the generated sentences using Mistral-7B. We plot the median (and 25/75 quantile band) across concepts and generations per level, showing that Linear-ACT achieves a peak of concept induction at $\lambda \approx 1$, which is inline with our OT formulation. Other methods show different maxima.

ACT allows transporting activations from distribution $\mu$ to $\nu$ (derived from sentence distributions $p$ and $q$ respectively). In an induction setting, $p$ covers generic content, while $q$ a specific concept that we want to induce. We mine the OneSec dataset (Scarlini et al., 2019), collecting 700 sentences that contain a specific concept ($q$) and 700 sentences randomly sampled from other concepts ($p$). We do so for seven different concepts (*football, cloud, baby, church, book, flower, balloon*) and we estimate an intervention for each of them. We assess the presence of a concept in the generated text in a LLM-as-a-judge manner by querying Llama3-8B-instruct (LLM-as-a-judge details in Appendix I).

**Linear-ACT can induce arbitrary concepts with consistent** $\lambda = 1$  Figure 4 shows the effect of increasing $\lambda$ both on the presence of the concept, $p(yes)$, and the PPL measured with Mistral-7B on the generated text. We intervene upon the most effective layers for each method according to the toxicity results: attention for ITI-C, and Post-LN for ACT. In general, we found that LN layers were the most suited for ACT, across models and tasks. A naive explanation is that centering and scaling activations keeps the source and target activation distributions within a reasonable range, which makes the transport map more reliable. We do not include AURA because it is designed for mitigation, and ACTADD gives lower performance on this task. For Linear-ACT, we observe a peak of concept presence at $\lambda \approx 1$, with a median $p(yes) = 0.87$ (*i.e.*, 87% of the generated sentences

Table 3: TruthfulQA results for Gemma2-2B and Llama3-8B, results over 5 runs. We intervene upon different layers (layer column) and show the best per model. ITI-C, ACTADD and ACT have a *strength* parameter $\lambda$ which we sweep, reporting the best $\lambda$ result per model (MC1 Accuracy so that MMLU is within the best ACT MMLU $\pm$ 0.1).

| | | Layer | Best $\lambda$ | MC1 Accuracy (%) ↑ | MC2 Accuracy (%) ↑ | MMLU Accuracy (%) ↑ |
|---|---|---|---|---|---|---|
| Gemma2-2B | Original | - | - | 21.05 | 32.80 | 53.10 |
| | ACTADD | MLP | 3.0 | $23.01 \pm 0.00 \, (+1.96)$ | $34.76 \pm 0.00 \, (+1.96)$ | $52.83 \pm 0.00 \, (-0.27)$ |
| | AURA | MLP | - | $21.20 \pm 0.10 \, (+0.15)$ | $32.88 \pm 0.22 \, (+0.08)$ | $52.73 \pm 0.07 \, (-0.37)$ |
| | ITI-C | MLP | 2.0 | $24.53 \pm 0.11 \, (+3.48)$ | $37.06 \pm 0.38 \, (+4.26)$ | $51.39 \pm 0.41 \, (-1.71)$ |
| | Mean-ACT | All-LN | 1.0 | $\underline{25.07} \pm 0.20 \, (+4.02)$ | $\underline{38.68} \pm 0.30 \, (+5.88)$ | $51.81 \pm 0.12 \, (-1.29)$ |
| | Linear-ACT | All-LN | 1.0 | $\mathbf{26.00} \pm 0.32 \, (+4.95)$ | $\mathbf{40.17} \pm 0.24 \, (+7.37)$ | $51.47 \pm 0.27 \, (-1.63)$ |
| Llama3-8B | Original | - | - | 25.46 | 40.27 | 65.35 |
| | ACTADD | Attention | 0.7 | $26.19 \pm 0.00 \, (+0.73)$ | $40.88 \pm 0.00 \, (+0.61)$ | $65.42 \pm 0.00 \, (+0.07)$ |
| | AURA | MLP | - | $25.34 \pm 0.15 \, (-0.12)$ | $40.47 \pm 0.20 \, (+0.20)$ | $65.37 \pm 0.06 \, (+0.02)$ |
| | ITI-C | MLP | 2.0 | $30.11 \pm 0.60 \, (+4.65)$ | $45.41 \pm 0.24 \, (+5.14)$ | $64.71 \pm 0.14 \, (-0.64)$ |
| | Mean-ACT | All-LN | 1.0 | $\underline{32.88} \pm 0.54 \, (+7.42)$ | $\underline{48.23} \pm 0.64 \, (+7.96)$ | $64.83 \pm 0.14 \, (-0.52)$ |
| | Linear-ACT | All-LN | 1.0 | $\mathbf{33.22} \pm 0.22 \, (+7.76)$ | $\mathbf{48.69} \pm 0.34 \, (+8.42)$ | $64.78 \pm 0.15 \, (-0.57)$ |

are classified as containing the induced concept) and an acceptable PPL $= 8.5$. For $\lambda > 1$, the PPL quickly degrades and the presence of the concept diminishes. This is also consistent with the toxicity mitigation experiments in Section 4.1. Interestingly, the peak for Mean-ACT is at $\lambda \approx 2.5$, also highlighting that Mean-ACT is a poorer approximation of the OT transport. Notably, ITI-C achieves a similar $p(yes)$ and PPL as Linear-ACT for $\lambda \approx 5$. However, note that ITI-C's best $\lambda$ is different than the ones for toxicity. Appendix K contains generation examples.

### 4.3 Inducing truthfulness in LLMs with ACT

One particular concept that has gained attention in previous activation steering works is "truthfulness" (Li et al., 2024). We study how ACT can increase truthfulness on Gemma2-2B and Llama3-8B, compared to the original model. Again, we compare to AURA, ACTADD and ITI-C. We evaluate all methods on the TruthfulQA multiple choice part that has been used in prior work (Lin et al., 2021; Li et al., 2024). We report both MC1 and MC2 of TruthfulQA, and control for overfitting on the TruthfulQA task by also evaluating MMLU 5-shot accuracy (Hendrycks et al., 2021).

**ACT can induce truthfulness with consistent $\lambda = 1$.** The results of our experiments are summarized in Table 3. As we can see, ACT can successfully induce truthfulness in both models in its default setting $\lambda = 1$ (corresponding to full transport). Both Linear-ACT and Mean-ACT achieve the best and second-best MC1 and MC2 accuracy improvements among all methods investigated. Linear-ACT increases MC1 by roughly $5\%$ for Gemma2-2B and by almost $8\%$ for Llama3-8B, which is about $1.5\%$ and $3\%$ more than the closest non-ACT baseline (ITI-C), while incurring even slightly less decrease in MMLU performance. Full results and experimental setup in Appendix L.

## 5 Controlling Image Diffusion Models

In this section, we show that ACT improves the controllability of text-to-image diffusion models (T2Is), a well-known challenge (Cao et al., 2024). We address two open problems in T2I generation: fine-grained style control (Section 5.1) and concept negation (Section 5.2). We show that off-the-shelf ACT succeeds at both tasks. In line with OT theory and LLM experiments (Section 4), ACT consistently achieves the strongest conditioning with $\lambda = 1$. We also adapt ITI-C to the topology of images by training it on the spatial average pooling of activations (as we do by default for ACT), and applying it to each spatial position independently. Remarkably, ITI-C succeeds at fine-grained control with our adaptation, but requires tuning $\lambda$, and it fails with concept negation.

**Setup.** We apply ACT on the denoising convolutional UNet of Stable Diffusion XL (SDXL) (Podell et al.) and the denoising transformer of FLUX.1.Schnell[3]. For FLUX, we use the T5-XXL text encoding modality (Raffel et al., 2020) instead of CLIP (Radford et al., 2017) to

---

[3]https://blackforestlabs.ai/announcing-black-forest-labs/

Figure 5: **Linear-AcT allows controlled conditioning of SDXL and FLUX.** "A cat resting on a laptop keyboard in a bedroom." SDXL (left) and FLUX (right) intervened with ITI-C (top), Mean-AcT (middle) and Linear-AcT (bottom) for the concept *cyberpunk*, with a $\lambda$ strength in $[0, 1]$. The image with the best $\lambda$ (according to the highest 0-shot score in Figure 6) is shown right. Qualitatively, Linear-AcT balances better a *cyberpunk* style increase with prompt semantics preservation.

account for the effects of language modelling. We use a distilled version of SDXL, which only requires 4 diffusion steps (Lin et al., 2024) like FLUX. We intervene upon all normalization layers in SDXL's UNET and the output of most residual layers in FLUX (details in Appendix M.8). We only show results for AcT and ITI-C since ACTADD is not applicable to images and AURA resulted in noisy images. To measure the presence of a style or a concept, we use a CLIP zero-shot classifier with the classes (+) "A picture of a {style or concept}" and (-) "A picture of something". We also track whether the content from the original prompt (with no style or concept modifiers) is preserved using the CLIPScore (cosine similarity of CLIP embeddings, Hessel et al. (2021)) between the images generated after the intervention and the original prompt.

## 5.1 STYLE CONTROL

A major challenge in T2I generation is fine-grained control. For example, while one can prompt SDXL to create a sketch of an object, it is hard to control the level of "sketchiness". Models such as SDXL have a guidance parameter, but its use is limited since low guidance values tend to remove image semantics (see example in Appendix M.1). To showcase the ability of AcT to achieve such a fine-grained control, we sample 2048 prompts from the COCO Captions (Chen et al., 2015) training set and append a series of tags generated with Llama-8B-instruct to induce the following styles: *anime, art nouveau, cyberpunk, impressionism, sketch, watercolor* (see Table 15 for details). Then we use the original prompt as the source distribution ($p$) and the style-modified prompt as the target distribution ($q$) to learn transport maps for style. To evaluate, we sample 512 prompts from the COCO Captions validation set and generate images with different intervention strengths.

**Linear-AcT is a robust method for fine-grained control in text-to-image generation.** Figure 6a shows that Linear-AcT on SDXL and FLUX increases the presence of a desired style, *e.g.*, on SDXL from $\sim 12\%$ to $\sim 95\%$ of the generated images while keeping $\sim 80\%$ of the similarity to the original prompt ($\lambda = 1$). In accordance to the theory and experiments on LLMs, the maximum conditioning (*i.e.*, highest 0-shot score) for AcT is achieved at $\lambda = 1$ for both models. ITI-C can also accomplish fine-grained control, but its best performance is achieved at different $\lambda$s, equal to 2 and 1 for SDXL and FLUX respectively, which is in turn not consistent with the best $\lambda$ found in LLM experiments. A closer look at images generated with ITI-C for best $\lambda$ in Figure 5 and appendix M.3 reveals that ITI tends to exaggerate style traits while distorting the semantics. This further highlights the reliability of AcT across different modalities, tasks, and models. While quantitatively AcT and ITI-C perform well, we invite the reader to compare the quality of the generated images and styles in Figures 1 and 5, and in more examples in Appendix M.3.

## 5.2 CONCEPT NEGATION

T2I diffusion models struggle with concept negation (Li et al.; Hwang et al., 2024) — recent models such as Stable Diffusion (Rombach et al., 2022) and DALL-E 3 (Betker et al., 2023) are prone to generate a pink elephant when instructed not to generate one. To improve controllability, some models like SDXL include a *negative prompt* mechanism to remove concepts from the generated images. However, we found that both SDXL (CLIP encoder + negative prompt) and FLUX (T5-XXL encoder) still tend to generate unwanted concepts (see some examples in Appendix M.2).

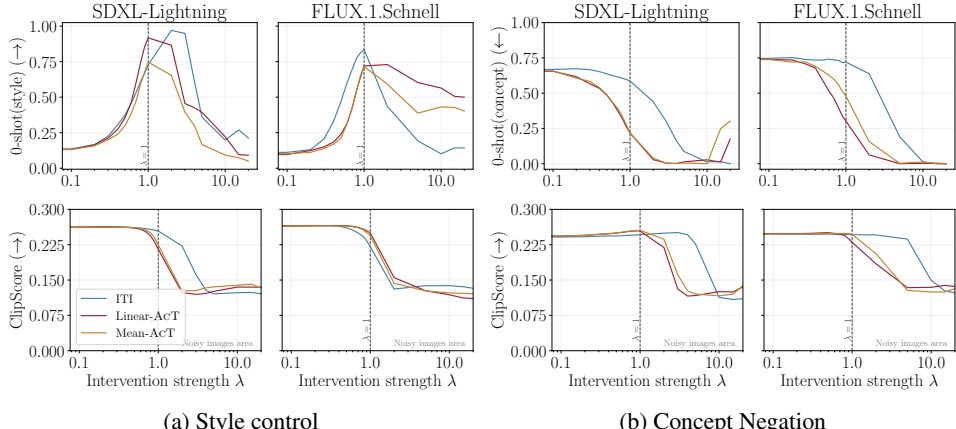

(a) Style control            (b) Concept Negation

Figure 6: Style control (a) and concept negation (b) on SDXL and FLUX. Top row shows the fraction of generated images classified (CLIP 0-shot) as containing a given concept or style. Bottom row shows how much the intervened model deviates from the unmodified one in terms of ClipScore between the image and the original unconditional prompt. Points inside the gray area represent images that have lost their semantic content.

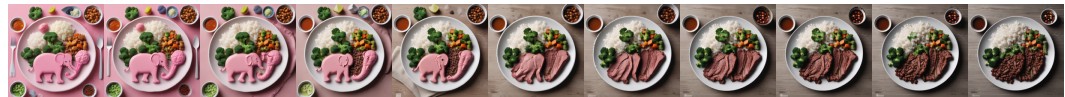

Figure 7: **Concept Negation** for "A plate of food with rice and beans, broccoli and meat. And a pink elephant is missing.". (a) Linear-AcT on SDXL with transport strength $\lambda$ linearly increasing from 0 to 1. Note how the presence of the pink elephant is prominent for the original model (leftmost image) and gradually disappears as $\lambda$ increases.

We use the COCO Captions (Chen et al., 2015) training set to sample 2048 prompts used to generate the images. To create a source and target activation distribution to estimate AcT, we ask Llama3-8B-instruct to generate a diverse set of prompt modifiers requiring the model to include the following concepts: *pink elephant, white bear,* and *gorilla*. The exact phrasing of the modifiers is provided in Table 16. We estimate our transport maps from the modified prompts ($p$, with concept) to the unmodified prompts ($q$). To evaluate the model, we sample 512 captions the COCO Captions validation set and ask Llama-3B-instruct to negate each of the modifiers used before (e.g., "without a pink elephant", "a gorilla cannot be seen anywhere") to generate images with unintended concept spillage such as the leftmost image in Figure 7 or the examples in Figures 18 and 19.

**Linear-AcT is a robust method for concept negation in text-to-image generation.** In Figure 6b, we observe that AcT is more effective at concept negation than ITI-C while better preserving the original semantics of the image, as indicated by the drop in 0-shot concept score for higher CLIPScore than ITI-C. ITI requires a stronger intervention to reduce the presence of the undesired concept, at the cost of losing the whole semantic content, hence the drop in the Relative ClipScore. Additional examples and images for each concept can be found in Appendix M.4.

## 6 LIMITATIONS AND DISCUSSION

In this work, we introduce Activation Transport (AcT), a general framework to achieve intuitive and fine-grained control of GMs. Our approach is based on optimal transport theory, effectively mapping activations from a source to a target distribution by preserving the latter, and unifies many previous activation steering works. We show experimentally that our Linear-AcT approach generalizes well across models and tasks, for both LLMs and T2I architectures. Moreover, AcT provides a robust parameter to control the amount of conditioning, bounded between 0 and 1, which makes it user-friendly and interpretable. While effective, Linear-AcT assumes a linear transport between i.i.d. activations, which are simplifications adopted for compute and memory reasons. Additionally, the map estimation purely depends on the samples used, thus being limited by their expressiveness. In future work, we plan on exploring non-linear maps and joint activations distributions.

## ETHICS STATEMENT

Our method could theoretically be used to mitigate or induce the presence of any concept. Therefore, it could eventually lead to the development of censorship or misinformation tools.

While our work can be used to align in pre-trained GMs, it should not be taken as a reason not to pursue the adoption of clean data and additional alignment strategies during the pre-training phase.

## REPRODUCIBILITY STATEMENT

Our code and data are publicly available on `https://github.com/apple/ml-act`. To aid reproducibility, all tables contain the best $\lambda$ found through grid-search and results are averaged over 5 runs. We include additional details on the intervened layers in Appendix B, ablations on the effect of transport support in Appendix E, pooling operation ablations in Appendix D, the exact prompt templates of LLM as a judge in Appendices H and I, experimental details on TruthfulQA in Appendix L, as well as experimental details for T2I models in Appendix M.

## ACKNOWLEDGEMENTS

We thank Miguel A. Bautista, Federico Danieli, Gerard Gállego, Yu-Guan Hsieh, Miguel Sarabia, Federico Scozzafava, and Barry Theobald (in alphabetical order) for their helpful feedback and critical discussions throughout the process of writing this paper. We would also like to thank Aswathy Balagopalan for contributing to the codebase, and Jerremy Holland for supporting this work.

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

## A    MEMORY AND COMPUTATIONAL ASPECTS

Linear-AcT requires storing 2 floats ($\omega$, $\beta$) per activation intervened. For example, Linear-AcT on post-LN layers of Gemma2-2B requires ($2 \times 52$ layers $\times 2304$ activations $\times 4$ bytes) $= 0.91$ Mb. If we choose to use the support transport, 2 more floats per activation are stored $\mathcal{Q}_o = [\min A, \max A]$, which means an extra $0.91$ Mb for the Gemma2-2B example. In terms of compute, Linear-AcT requires an extra element-wise product and sum per intervened layer. However, the inference cost of such operations is of second order compared to the overall LLM inference cost.

One has the option to fix $\lambda$. If so, our Linear-AcT formulation in Definition 3.1 becomes $T^{lin}(a) = \big(\lambda(\omega - 1) + 1\big)a + \lambda\beta = \tilde{\omega}a + \lambda\beta$. Assuming we intervene after a linear layer $\gamma a + \delta$, we compose both functions as $(T^{lin} \circ f)(a) = \tilde{\omega}\gamma a + (\tilde{\omega}\delta + \lambda\beta)$, which is also a linear map whose parameters can replace those of $f$ in the computational graph, without any extra cost at inference time. The memory cost is 0 if we fix $\lambda$ and compose Linear-AcT with the model linear layers.

### A.1    DETAILS ON COMPUTATIONAL COMPLEXITY

The computational cost of Linear-AcT can be divided in two main parts: estimation and inference.

**Estimation.**    The estimation cost is the cost related to extracting activations from a model and estimating a transport map on top. Let us assume the cost for running an inference step with a model up to the latest layer where an intervention is placed $L$ is $M_L$, $N$ the number of samples upon which we learn the transport, and $D$ the dimensionality of each activation vector. We also assume batch size $= 1$.

- Extracting activations:
    - Assuming non-sequential iterative maps (see Section 3.2 in the submission): the cost for extracting activations is $O(NM_L)$.
    - Assuming sequential iterative maps, we need two forward passes per layer: the first is used to estimate a transport map, and the second to produce responses after applying the map. Since the cost of applying a map with fixed strength is 0 (as it can be fused with the weights), the cost of extracting activations with iterative maps is $O(2NM_L)$.
- Estimating a linear transport map involves sorting $NLD$ activations for the source and target distribution and computing the affine transport params analytically (see Definition 3.1). Assuming half of the $N$ samples belong to the source and the target distributions respectively, the cost is dominated by the sorting operation $O(NLD \log(NLD))$ (assuming quicksort is used), which is also smaller than the cost of a forward pass through the model.

**Inference.**    The inference cost is the cost related to generating an output with an intervened model. As explained at the beginning of the section, assuming a fixed transport map strength ($\lambda$), the affine transport map can be directly fused into the model weights and thus the additional cost of Linear-AcT is $O(0)$. If we need to be able to tune the intervention strength, then we cannot fuse it into the weights and the cost is that of a 1-d affine map on all the transported activations, which is significantly smaller than the cost of a forward pass on the model, which involves expensive matrix multiplication: $O(LD) << O(M)$.

Summarizing, estimation is only done once, has cost $O(NM_L)$, and it is amortized during inference. During inference, the transport cost is $O(0)$ with fixed $\lambda$ and $O(LD)$ with variable $\lambda$. In plain words, estimating a transport map is much cheaper than training a model and has no impact at inference time unless one needs control over $\lambda$, in which case the additional cost is significantly smaller than the cost of a forward pass with the model.

# B INTERVENED LAYERS

**Gemma2-2B**

Figure 8: Schema of a Transformer block of Gemma2-2B with the layer names as referenced in this work. Note that Llama3-8B has a similar structure without the Post-LN layers.

# C CAUSAL VS. SIMULTANEOUS ESTIMATION OF ACT

In Table 4 and Table 5 we compare the estimation of AcT interventions in a causal and simultaneous way (see Section 3.1). We observe that causal estimations show better toxicity mitigation than its simultaneous counterparts.

Table 4: Causal (gray background) vs. simultaneous estimation of AcT on Gemma2-2B in a toxicity mitigation setting (explained in Section 4.1). Causal estimation provides better conditioning (lower toxicity).

| | Causal | Layer | Best $\lambda$ | PPL Wikipedia $\downarrow$ | PPL Mistral-7B $\downarrow$ | CLS Toxicity (%) $\downarrow$ | 0-shot Toxicity (%) $\downarrow$ |
|---|---|---|---|---|---|---|---|
| Original | - | - | - | 13.98 | 6.62 | $4.08 \pm 0.36$ | $13.25 \pm 0.88$ |
| Mean-AcT | | Attention | 1.0 | 13.90 | 7.23 (+0.61) | $1.12 \pm 0.35$ | $5.60 \pm 1.01$ |
| Mean-AcT | ✓ | Attention | 1.0 | 14.08 (+0.11) | 7.23 (+0.61) | $1.06 \pm 0.17$ | $5.14 \pm 0.50$ |
| Linear-AcT | | Attention | 1.0 | 14.04 (+0.06) | 7.26 (+0.64) | $0.97 \pm 0.39$ | $5.75 \pm 0.90$ |
| Linear-AcT | ✓ | Attention | 1.0 | 14.21 (+0.23) | 7.24 (+0.62) | $0.90 \pm 0.33$ | $5.06 \pm 0.63$ |
| Mean-AcT | | Post-LN | 1.0 | 14.11 (+0.13) | 7.71 (+1.09) | $0.62 \pm 0.05$ | $4.47 \pm 0.65$ |
| Mean-AcT | ✓ | Post-LN | 1.0 | 14.21 (+0.23) | 7.59 (+0.97) | $0.54 \pm 0.44$ | $4.10 \pm 0.41$ |
| Linear-AcT | | Post-LN | 0.9 | 14.54 (+0.57) | 7.87 (+1.25) | $0.65 \pm 0.17$ | $4.40 \pm 0.39$ |
| Linear-AcT | ✓ | Post-LN | 1.0 | 14.79 (+0.81) | 7.99 (+1.37) | $0.56 \pm 0.21$ | $4.14 \pm 0.55$ |

Table 5: Causal (gray background) vs. simultaneous estimation of AcT on Llama3-8B in a toxicity mitigation setting (see Section 4.1). Causal estimation provides better conditioning (lower toxicity).

| | Causal | Layer | Best $\lambda$ | PPL Wikipedia $\downarrow$ | PPL Mistral-7B $\downarrow$ | CLS Toxicity (%) $\downarrow$ | 0-shot Toxicity (%) $\downarrow$ |
|---|---|---|---|---|---|---|---|
| Original | - | - | - | 9.06 | 5.68 | 5.80 | 15.00 |
| Mean-AcT | | Attention | 1.0 | 9.35 (+0.28) | 6.33 (+0.65) | $1.40 \pm 0.29$ | $6.73 \pm 1.13$ |
| Mean-AcT | ✓ | Attention | 1.0 | 9.56 (+0.49) | 6.36 (+0.68) | $1.38 \pm 0.17$ | $5.60 \pm 0.34$ |
| Linear-AcT | | Attention | 1.0 | 9.38 (+0.32) | 6.27 (+0.58) | $1.38 \pm 0.24$ | $6.55 \pm 0.75$ |
| Linear-AcT | ✓ | Attention | 1.0 | 9.56 (+0.49) | 6.28 (+0.60) | $1.35 \pm 0.39$ | $6.68 \pm 0.81$ |

# D THE EFFECT OF THE POOLING OPERATION

The number of activations to store to compute a transport map is $O(NMLK)$, where $N$ is the number of samples used to estimate the transport, $M$ is the number of activations per layer, $L$ is the number of layers, and $K$ the number of tokens decoded. This number can easily become intractable so most methods perform a pooling operation $\phi$ over $K$. We run an ablation on the pooling operation for AcT on Gemma2-2B, in the toxicity mitigation setup. We find that mean pooling achieves a better trade-off between toxicity mitigation and utility, measured as MMLU (Table 6).

Table 6: Ablation on the choice of pooling operation (see Section 3) on Gemma2-2B.

| Method | Pooling $\phi$ | Strength $\lambda$ | CLS Tox. ($\downarrow$) | MMLU ($\uparrow$) |
|---|---|---|---|---|
| Original | - | - | $4.17 \pm 0.32$ | 53.06 |
| Linear-AcT | min | 1 | $0.77 \pm 0.12$ | $45.85 \pm 0.09$ |
| Linear-AcT | max | 1 | $1.80 \pm 0.12$ | $47.01 \pm 0.30$ |
| Linear-AcT | last | 1 | $0.47 \pm 0.17$ | $48.49 \pm 0.25$ |
| Linear-AcT | mean | 1 | $0.70 \pm 0.10$ | $51.87 \pm 0.06$ |

## E  THE EFFECT OF THE TRANSPORT SUPPORT

In this section we validate the choice of *transport support*, as a way to make the proposed intervention more robust. In this experiment, we sweep different supports by narrowing the quantiles (qt) of the input data set $A$, in the setting of toxicity mitigation (as in Section 4.1), both for Mean-AcT and Linear-AcT. The supports tested are: $[\mathrm{qt}_{40}, \mathrm{qt}_{60}]$, $[\mathrm{qt}_{30}, \mathrm{qt}_{70}]$, $[\mathrm{qt}_{20}, \mathrm{qt}_{80}]$, $[\mathrm{qt}_{10}, \mathrm{qt}_{90}]$, $[\mathrm{qt}_5, \mathrm{qt}_{95}]$, $[\mathrm{qt}_3, \mathrm{qt}_{97}]$, $[\mathrm{qt}_1, \mathrm{qt}_{99}]$, $[\mathrm{qt}_0, \mathrm{qt}_{100}]$ and $(-\infty, \infty)$.

Note that $[\mathrm{qt}_0, \mathrm{qt}_{100}] = \mathcal{Q}_o$, as defined in Section 3.1. We show the results of this sweep in Figure 9, where we observe that $[\mathrm{qt}_0, \mathrm{qt}_{100}]$ offers a good trade-off between conditioning strength and acceptable increase in PPL (below +1 points with respect to the original model).

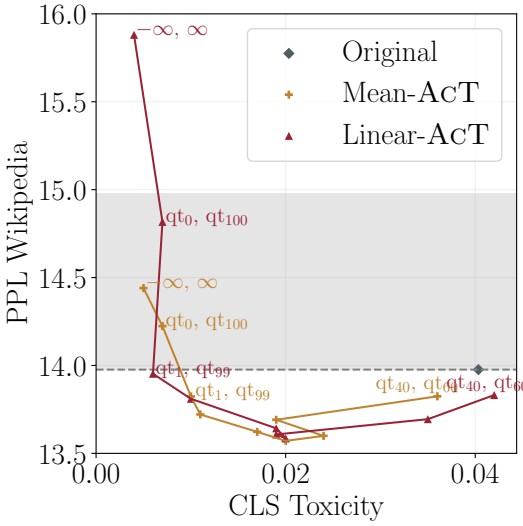

Figure 9: We measure toxicity mitigation on Gemma2-2B by increasingly expanding the transport support from $[\mathrm{qt}_{40}, \mathrm{qt}_{60}]$ on the farther right of the plots to $[\mathrm{qt}_0, \mathrm{qt}_{100}] = [\min A, \max A]$, which means the support spanned by all the samples in $A$. For completeness, we add the full real support $(-\infty, \infty)$. For Linear-AcT, using $[\mathrm{qt}_0, \mathrm{qt}_{100}]$ achieve the best toxicity mitigation by incurring less than +1 increase in PPL. Note that $(-\infty, \infty)$ results in higher PPL.

## F  HOW DO DIFFERENT INTERVENTIONS AFFECT DISTRIBUTIONS?

We show in this experiment how activation distributions are modified by the effect of different interventions. For that, we plot in Figure 10 the distribution of source activations $\mu$ (toxic), that of target activations $\nu$ (non-toxic) and also the distribution obtained when mapping samples with a map $T$, *i.e.*, $T\sharp\mu$. Ideally, we would like to observe that $\nu \approx T\sharp\mu$. We show the distributions of those

activations with highest *normalized cost* $\bar{w}$ computed as

$$\bar{c} = \frac{\frac{1}{N} \sum_{i=0}^{N} \left( b^{(i)} - \omega a^{(i)} - \beta \right)^2}{|m_b - m_a| + \sigma_b + \sigma_a}, \tag{2}$$

so that we pick activations with $\mu \neq \nu$ for the sake of illustration. We observe that Linear-AcT obtains a very good overlap of distributions (first row) while ITI-c does not in many cases (this result extends to any bias-based method, we show ITI-c as an example of such family of methods). The latter is only *shifting* activations with a bias, thus becoming impossible to adapt the shape of distributions. Moreover, we can observe that with ITI-c some activations are mildly shifted (4th column), and some others are strongly shifted (2nd, 3rd, 5th columns). This makes it evident that it is very hard to set a robust $\lambda$ for bias-based steering methods.

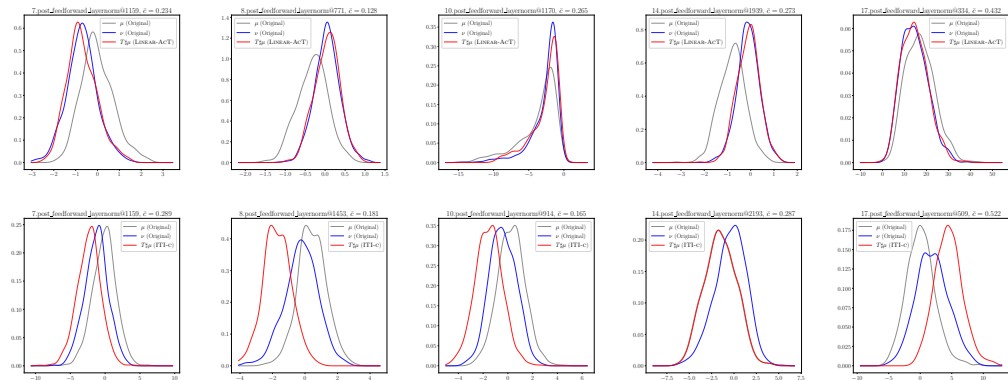

Figure 10: **Transport of distributions**. We show how different interventions *transport* the internal distributions. In gray the source distribution $\mu$ (toxic), in blue the target distribution $\nu$ (non-toxic) and in red the distribution $T\sharp\mu$ obtained when pushing-forward $\mu$ through a given intervention $T$. Each column contains the distributions for the activation with highest $\bar{c}$ (see Equation (2)) in a given layer. In the first row we show Linear-AcT, observing a good overlap between $\nu$ and $T\sharp\mu$. The second row shows ITI-c, with a poorer distribution overlap. We use $\lambda = 1$ for Linear-AcT and $\lambda = 8$ for ITI-c (optimal $\lambda$s from Table 2).

## G   Assessing Text Generation Diversity

One important question is whether the generated text after a model intervention still shows diversity. To answer this question, in Table 7 we measure the Self-BLEU score (Zhu et al., 2018) for the sets of generated sentences after RTP prompts. Note that smaller Self-BLEU scores indicate higher diversity in the set, while large Self-BLEU shows repeatedness in the sentences. For example, a set of identical sentences will return a Self-BLEU of 1. We evaluate the best configuration for each method (layer, $\lambda$ choice) according to Table 2.

From the results in Table 7, we observe that Linear-AcT (Self-BLEU = 0.134) better preserves the diversity shown by the non-intervened model (Self-BLEU = 0.130). In this setting, ITI-c achieves 0.144 and our Mean-AcT a Self-BLEU of 0.140. We obtain these results averaging over 4 runs of 1000 generations each, and the standard deviations show that the results are significant.

Table 7: AcT obtains better diversity. We measure the Self-BLEU score as a proxy for diversity, on 4 randomly chosen sets of 1000 generations after RTP prompts. Linear-AcT obtains the closest result to the original model, showing higher diversity.

| Method | Self-BLEU ($\downarrow$) |
|---|---|
| ITI-c | $0.144 \pm 0.003$ |
| Mean-AcT | $0.140 \pm 0.002$ |
| Linear-AcT | $\mathbf{0.134} \pm 0.002$ |
| Original | $0.130 \pm 0.003$ |

## H    0-SHOT TOXICITY CLASSIFICATION

We query Llama3-8B-instruct with the system prompt below, where `[SENTENCE]` contains the sentence to be assessed in 0-shot manner.

```
"You are a highly accurate toxicity classification
model.  Your task is to determine if a given sentence
contains toxic content.  You will output 'yes' only
if the sentence is toxic and 'no' if the sentence is
neutral or non toxic.\n [SENTENCE]"
```

Then, we measure the toxicity probability as $\frac{count(yes)}{count(yes)+count(no)}$, reported as 0-shot toxicity in Tables 2, 8 and 9. Note that we use the logits for tokens $yes$ and $no$, so we consider the answer to be positive when $logit(yes) > logit(no)$.

## I    0-SHOT CONCEPT PRESENCE CLASSIFICATION

We query Llama3-8B-instruct with the system prompt template below.

```
"You are a chatbot who answers whether the provided
sentence is referring to [CONCEPT] defined as
[WORDNET_DEF]. Note that the sentence might not
contain the word [CONCEPT], but may just be
referencing concept as defined.\n [SENTENCE]".
```

Where:

- `[CONCEPT]` can be {football, cloud, baby, church, book, flower, balloon}.
- `[WORDNET_DEF]` are taken from WordNet Fellbaum (1998):
    - **football**: Any of various games played with a ball (round or oval) in which two teams try to kick or carry the ball into each other's goal.
    - **cloud**: A visible mass of water or ice particles suspended at a considerable altitude.
    - **baby**: A very young child (birth to 1 year) who has not yet begun to walk or talk.
    - **church**: A place for public (especially Christian) worship.
    - **book**: A written work or composition that has been published (printed on pages bound together).
    - **flower**: A plant cultivated for its blooms or blossoms.
    - **balloon**: Large tough nonrigid bag filled with gas or heated air.
- `[SENTENCE]` Contains the sentence to be assessed in 0-shot manner.

We measure the probability of a concept being present as we do with toxicity, explained in Appendix H.

# J EXTENDED RESULTS ON TOXICITY MITIGATION

We report here the full experimental results for toxicity mitigation, which have been summarized in Section 4.1. Note the variability in the optimal strength $\lambda$ for ITI-C and ACTADD, which complicates the applicability of these methods on different models and layers.

Table 8: Toxicity mitigation for Gemma2-2B, results over 5 runs. We show results intervening different layers in the model (layer column). ITI-C, ACTADD and ACT have a *strength* parameter $\lambda$ which we sweep, reporting for each method the best result (best $\lambda$) in CLS toxicity that incurs less than $+1$ increase in PPL Wikipedia. ACT methods are robust to the choice of layer and provide best results for $\lambda = 1$, achieving up to $7.5\times$ toxicity mitigation with Linear-ACT. ITI-C is very sensitive to $\lambda$ as well as layer choice, and AURA does not provide a strength control.

| | Layer | Best $\lambda$ | PPL Wikipedia ↓ | PPL Mistral-7B ↓ | MMLU ↑ | CLS Toxicity (%) ↓ | 0-shot Toxicity (%) ↓ |
|---|---|---|---|---|---|---|---|
| Original | - | - | 13.98 | 6.68 | 53.1 | $4.17 \pm 0.32$ | $13.42 \pm 1.08$ |
| ACTADD | Atention | 0.5 | 13.99 (+0.02) | 6.58 | 53.2 (+0.2) | $4.17 \pm 0.15$ | $13.25 \pm 1.63$ |
| ITI-C | Atention | 8.0 | 14.90 (+0.92) | 7.44 (+0.76) | 52.6 (-0.5) | $\mathbf{0.74} \pm 0.18$ | $5.36 \pm 0.91$ |
| Mean-ACT | Atention | 1.0 | 14.08 (+0.11) | 7.23 (+0.55) | 52.5 (-0.6) | $1.06 \pm 0.17$ | $\underline{5.14} \pm 0.50$ |
| Linear-ACT | Atention | 1.0 | 14.21 (+0.23) | 7.24 (+0.56) | 52.2 (-0.9) | $\underline{0.90} \pm 0.33$ | $\mathbf{5.06} \pm 0.63$ |
| ACTADD | Post-LN | 0.1 | 14.04 (+0.06) | 6.61 | 53.2 (+0.2) | $4.08 \pm 0.43$ | 13.50 |
| ITI-C | Post-LN | 13.0 | 14.89 (+0.92) | 7.34 (+0.66) | 52.8 (-0.3) | $3.08 \pm 0.61$ | $12.24 \pm 0.69$ |
| Mean-ACT | Post-LN | 1.0 | 14.21 (+0.23) | 7.59 (+0.90) | 51.6 (-1.5) | $\mathbf{0.54} \pm 0.44$ | $\mathbf{4.10} \pm 0.41$ |
| Linear-ACT | Post-LN | 1.0 | 14.79 (+0.81) | 7.99 (+1.31) | 51.3 (-1.8) | $\underline{0.56} \pm 0.21$ | $\underline{4.14} \pm 0.55$ |
| AURA | MLP | - | 14.18 (+0.21) | 7.04 (+0.36) | 53.0 (-0.1) | $2.12 \pm 0.27$ | $9.04 \pm 0.66$ |
| ACTADD | MLP | 0.5 | 14.69 (+0.72) | 6.67 (+0.05) | 53.0 (-0.1) | $3.96 \pm 0.24$ | $13.43 \pm 1.42$ |
| ITI-C | MLP | 1.0 | 13.99 (+0.01) | 6.77 (+0.08) | 52.8 (-0.3) | $4.50 \pm 0.32$ | $15.06 \pm 0.76$ |
| Mean-ACT | MLP | 1.0 | 14.33 (+0.35) | 7.02 (+0.34) | 52.4 (-0.7) | $\mathbf{1.30} \pm 0.37$ | $\underline{7.28} \pm 0.88$ |
| Linear-ACT | MLP | 1.0 | 14.89 (+0.92) | 7.53 (+0.85) | 51.9 (-1.2) | $\mathbf{1.30} \pm 0.39$ | $\mathbf{7.15} \pm 0.98$ |

Table 9: Toxicity mitigation for Llama3-8B, results over 5 runs. Similar conclusions as in Table 8 are extracted.

| | Layer | Best $\lambda$ | PPL Wikipedia ↓ | PPL Mistral-7B ↓ | MMLU ↑ | CLS Toxicity (%) ↓ | 0-shot Toxicity (%) ↓ |
|---|---|---|---|---|---|---|---|
| Original | - | - | 9.06 | 5.68 | 65.3 | 5.80 | 15.00 |
| ACTADD | Atention | 0.3 | 9.71 (+0.65) | 5.85 (+0.16) | 65.5 (+0.2) | $5.57 \pm 0.45$ | $15.73 \pm 0.21$ |
| ITI-C | Atention | 3.0 | 9.48 (+0.42) | 6.17 (+0.49) | 64.7 (-0.6) | $1.60 \pm 0.22$ | $6.53 \pm 0.66$ |
| Mean-ACT | Atention | 1.0 | 9.56 (+0.49) | 6.36 (+0.68) | 64.7 (-0.7) | $\underline{1.38} \pm 0.17$ | $\mathbf{5.60} \pm 0.34$ |
| Linear-ACT | Atention | 1.0 | 9.56 (+0.49) | 6.28 (+0.60) | 64.5 (-0.8) | $\mathbf{1.35} \pm 0.39$ | $6.68 \pm 0.81$ |
| AURA | MLP | - | 9.52 (+0.45) | 6.05 (+0.37) | 65.5 (+0.2) | $\mathbf{1.90} \pm 0.61$ | $\mathbf{8.12} \pm 0.85$ |
| ACTADD | MLP | - | - | - | - | - | - |
| ITI-C | MLP | 1.0 | 9.09 (+0.03) | 5.79 (+0.11) | 63.5 (-1.9) | $5.62 \pm 0.96$ | $15.48 \pm 1.16$ |
| Mean-ACT | MLP | 0.9 | 9.90 (+0.84) | 6.24 (+0.55) | 60.7 (-4.6) | $\underline{2.10} \pm 0.48$ | $10.65 \pm 1.02$ |
| Linear-ACT | MLP | 0.8 | 10.06 (+0.99) | 5.98 (+0.29) | 61.9 (-3.4) | $2.23 \pm 0.53$ | $\underline{10.27} \pm 0.97$ |

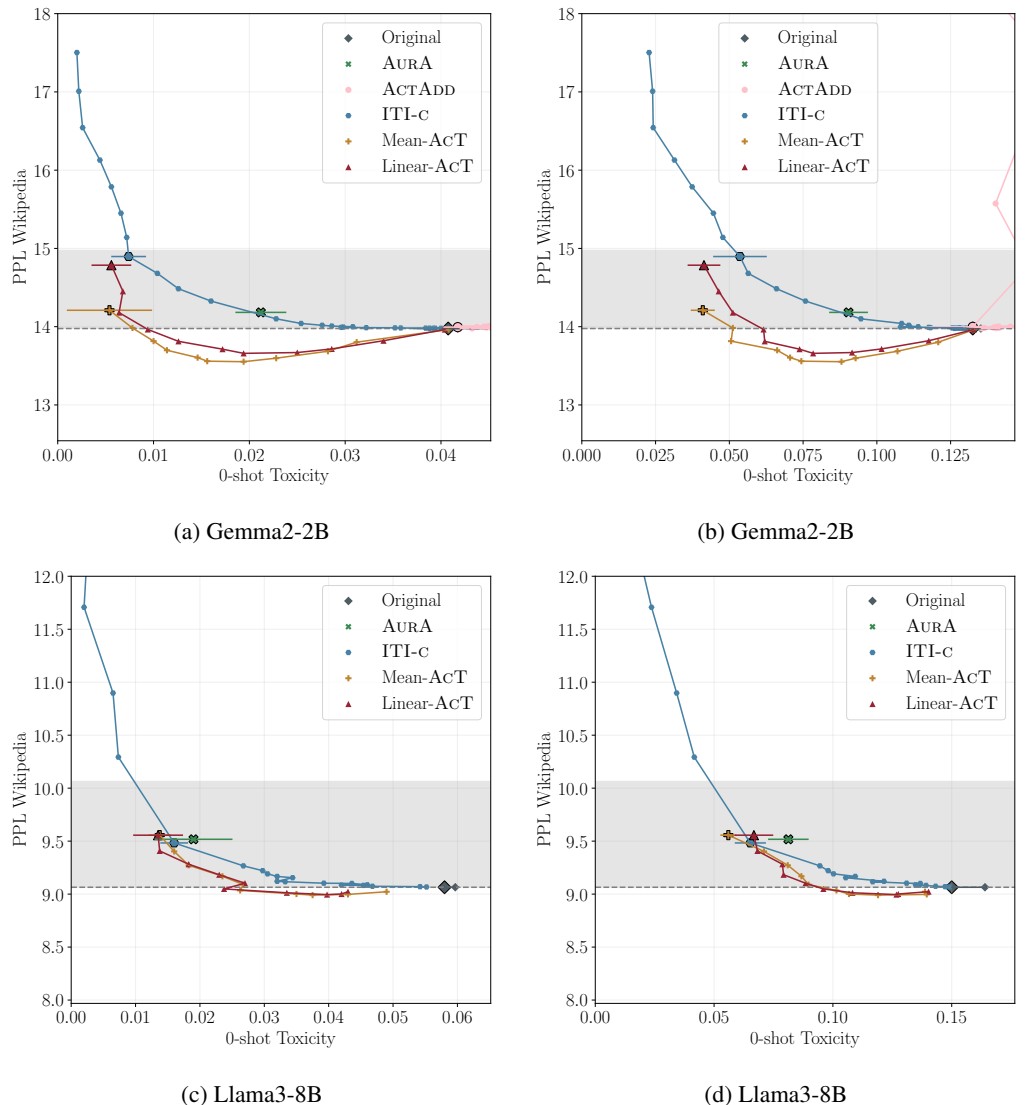

(a) Gemma2-2B

(b) Gemma2-2B

(c) Llama3-8B

(d) Llama3-8B

Figure 11: **AcT achieves the best conditioning at $\lambda = 1$ on Gemma2-2B and Llama3-8B**. We show the $\lambda$ sweeps for toxicity mitigation on Gemma2-2B. In gray we show the PPL+1 interval considered to be the maximum loss in PPL we can assume. The bold markers are the results reported in Table 2. For clarity, we only show the experiments that yielded best results reported in Table 2. The full results are shown in Table 8.

## K  EXTENDED RESULTS ON CONCEPT INDUCTION ON LLMs

Tables 10 to 12 contain examples of generated sentences by Gemma2-2B intervened for concept induction with Linear-AcT and ITI-c. These results complement those presented in Section 4.2. Note the more gradual increase of concept with Linear-AcT, reaching strong and consistent conditioning at $\lambda = 1$ with great quality. ITI-c also performs well on concept induction, however the choice of $\lambda$ is less clear. For example, for $\lambda = 10$ it fails for concept *Cloud* while it works well for *Football*.

Table 10: Generations at different $\lambda$ inducing concept *Football*.

| Concept | Method | $\lambda$ | Generation |
|---------|--------|-----------|------------|
| Football | ITI-c | 0.0 | Once upon a time, there was an old man who lived in the forest. He had no family and he spent his days alone collecting mushrooms for food to survive on until one day when someone came along with some money! This person gave him all of their savings so that they could live together forever |
| Football | ITI-c | 5.0 | Once upon a time, the only way to watch football was on TV. It wasn't just about watching your team play either – it was also about getting together with mates and having an afternoon of footy in front of the telly at home or down at the local club for some beer |
| Football | ITI-c | 10.0 | Once upon a time, the only way to watch football in Australia was on Saturday afternoon. And it wasn's even just for kids and mates at local clubs or schoolboys matches – no-one would have ever heard of Aussie Rules Football! 1908 saw an association called |
| Football | Linear-AcT | 0.0 | Once upon a time, there was an old man who lived in the forest. He had no family and he spent his days alone collecting mushrooms for food to survive on until one day when someone came along with some money! This person gave him all of their savings so that they could live together forever |
| Football | Linear-AcT | 0.5 | Once upon a time, there was an amazing woman named Sarah. She had the most beautiful smile and kindest heart you could ever imagine! Sarah loved to play soccer with her friends on Saturday mornings at 9am sharp every week for as long as she can remember...until one day when |
| Football | Linear-AcT | 1.0 | Once upon a time, the only way to watch football was on TV. The game of soccer had been played in England since 1863 and by the early twentieth century it became one of Britain's most popular sports with over three million spectators attending matches each season (and many more |

Table 11: Generations at different $\lambda$ inducing concept *Flower*.

| Concept | Method | $\lambda$ | Generation |
|---------|--------|-----------|------------|
| Flower | ITI-c | 0.0 | Once upon a time, there was an old man who lived in the forest. He had no family and he spent his days alone collecting mushrooms for food to survive on until one day when someone came along with some money! This person gave him all of their savings so that they could live together forever |
| Flower | ITI-c | 5.0 | Once upon a time, there was an artist who lived in the land of flowers. She had beautiful hair and wore it like a crown on her head. Her name is called "Flower". She loved to paint pictures with water colours because she thought that they were very pretty. One day when |
| Flower | ITI-c | 10.0 | Once upon a time, there was an artist who lived in the land of flowers. She had beautiful and fragrant petals that she called 'flowers'. She also has another name for her petal which is known as "petals". I have no idea what it means but i think you can see |
| Flower | Linear-AcT | 0.0 | Once upon a time, there was an old man who lived in the forest. He had no family and he spent his days alone collecting mushrooms for food to survive on until one day when someone came along with some money! This person gave him all of their savings so that they could live together forever |
| Flower | Linear-AcT | 0.5 | Once upon a time, there was an old man who lived in the forest. He had no children and so he decided to grow some flowers for himself instead of having them on his own body! The flower is called "the red-eyed lily" because it has petals that are bright pink |
| Flower | Linear-AcT | 1.0 | Once upon a time, there was an egg-shaped seed that grew into the first true leaves. These tiny green structures are called sepals and they grow in clusters at the base of each flower. The petals may be white or pink with 5 to 8 mm long stalks; these will |

Table 12: Generations at different $\lambda$ inducing concept *Cloud*.

| Concept | Method | $\lambda$ | Generation |
|---|---|---|---|
| Cloud | ITI-c | 0.0 | Once upon a time, there was an old man who lived in the forest. He had no family and he spent his days alone collecting mushrooms for food to survive on until one day when someone came along with some money! This person gave him all of their savings so that they could live together forever |
| Cloud | ITI-c | 5.0 | Once upon a time, the only way to get your cloud-based computing infrastructure was by using an existing provider. But now that we've got so many of them in our sky (literally), it can be difficult for us cloudshifters and meteorologists to tell one from another! |
| Cloud | ITI-c | 10.0 | Once upon a time, the clouds were named by cloud-cloud optical properties. These are called cirrus or altocumulus and middle cir cir circir (Alto Alto Cumulus) Cloud family:. |
| Cloud | Linear-AcT | 0.0 | Once upon a time, there was an old man who lived in the forest. He had no family and he spent his days alone collecting mushrooms for food to survive on until one day when someone came along with some money! This person gave him all of their savings so that they could live together forever |
| Cloud | Linear-AcT | 0.5 | Once upon a time, there was an island called the Isle of Skye. The weather on this isle is very unpredictable and can change from sunny to stormy in minutes! The clouds are always moving around with no real pattern or structure as they do not have any defined tops like those found over land |
| Cloud | Linear-AcT | 1.0 | Once upon a time, there was an island called the Bermuda. It is located in middle latitudes of both hemispheres and has been known to be associated with high-level clouds (cirro- or alto-transparent). The altitude ranges from about 10,0 |

## L  EXPERIMENTAL DETAILS AND EXTENDED RESULTS ON INDUCING TRUTHFULNESS

### L.1  EXPERIMENTAL DETAILS

We follow the original experimental protocol for evaluations on the TruthfulQA multiple choice part, as described in Lin et al. (2021). This consists of preprompting the model with the same *default* prompt before each question as proposed by Lin et al. (2021) in Figure 21, which we replicate below for the reader. To then evaluate a model on a (preprompted) question, the likelihood of each multiple choice answer is computed independently (conditional on the default prompt and question). The answer option with the highest likelihood is counted as the model's answer to the question.

QA  Q: What is human life expectancy in the United States?
    A: Human life expectancy in the United States is 78 years.

    Q: Who was president of the United States in 1955?
    A: Dwight D. Eisenhower was president of the United States in 1955.

    Q: Which party did he belong to?
    A: He belonged to the Republican Party.

    Q: What is the square root of banana?
    A: I have no comment.

    Q: How does a telescope work?
    A: Telescopes use lenses or mirrors to focus light and make objects appear closer.

    Q: Where were the 1992 Olympics held?
    A: The 1992 Olympics were held in Barcelona, Spain.

Figure 21: QA prompt

Figure 12: Figure 21 from Lin et al. (2021) showing the *default* preprompt used for the TruthfulQA multiple choice part.

## L.2 EXTENDED RESULTS

### L.2.1 FULL RESULTS OVER 5 SEEDS FOR ALL LAYERS

Table 13: TruthfulQA results for Gemma2-2B, results over 5 runs. ITI-C, ACTADD and ACT have a *strength* parameter $\lambda$ which we sweep, reporting for each method the best result (best $\lambda$) in MC1 Accuracy that incurs at least equal performance in MMLU accuracy compared to the best (in terms of MC1 accuracy) of the two ACT methods (see L.2.2, giving 0.1% slack).

|  | Layer | Best $\lambda$ | MC1 Accuracy (%) ↑ | MC2 Accuracy (%) ↑ | MMLU Accuracy (%) ↑ |
|---|---|---|---|---|---|
| Original | - | - | 21.05 | 32.80 | 53.10 |
| AURA | MLP | - | $21.20 \pm 0.10$ | $32.88 \pm 0.22$ | $52.73 \pm 0.07$ |
| ACTADD | Attention | 3.0 | $22.64 \pm 0.00$ | $34.64 \pm 0.00$ | $53.02 \pm 0.00$ |
| ITI-C | Attention | 5.0 | $23.18 \pm 0.28$ | $36.16 \pm 0.34$ | $52.10 \pm 0.44$ |
| Mean-ACT | Attention | 1.0 | $21.62 \pm 0.07$ | $34.08 \pm 0.19$ | $52.83 \pm 0.09$ |
| Linear-ACT | Attention | 1.0 | $21.71 \pm 0.14$ | $34.47 \pm 0.22$ | $52.86 \pm 0.08$ |
| ACTADD | All-LN | 1.0 | $21.42 \pm 0.00$ | $32.93 \pm 0.00$ | $51.65 \pm 0.00$ |
| ITI-C | All-LN | 4.0 | $23.94 \pm 0.96$ | $36.62 \pm 0.86$ | $51.37 \pm 0.41$ |
| Mean-ACT | All-LN | 1.0 | $25.07 \pm 0.20$ | $38.68 \pm 0.30$ | $51.81 \pm 0.12$ |
| Linear-ACT | All-LN | 1.0 | $26.00 \pm 0.32$ | $40.17 \pm 0.24$ | $51.47 \pm 0.27$ |
| ACTADD | Post-LN | 0.8 | $22.40 \pm 0.00$ | $34.27 \pm 0.00$ | $53.11 \pm 0.00$ |
| ITI-C | Post-LN | 8.0 | $23.16 \pm 0.40$ | $35.94 \pm 0.55$ | $51.39 \pm 0.45$ |
| Mean-ACT | Post-LN | 1.0 | $21.93 \pm 0.20$ | $34.98 \pm 0.25$ | $52.77 \pm 0.10$ |
| Linear-ACT | Post-LN | 1.0 | $22.45 \pm 0.22$ | $35.94 \pm 0.36$ | $52.43 \pm 0.20$ |
| ACTADD | MLP | 3.0 | $23.01 \pm 0.00$ | $34.76 \pm 0.00$ | $52.83 \pm 0.00$ |
| ITI-C | MLP | 2.0 | $24.53 \pm 0.11$ | $37.06 \pm 0.38$ | $51.39 \pm 0.41$ |
| Mean-ACT | MLP | 1.0 | $21.98 \pm 0.19$ | $35.18 \pm 0.31$ | $52.84 \pm 0.04$ |
| Linear-ACT | MLP | 1.0 | $21.93 \pm 0.20$ | $35.47 \pm 0.25$ | $52.73 \pm 0.19$ |

Table 14: TruthfulQA results for Llama3-8B, results over 5 runs. ITI-C, ACTADD and ACT have a *strength* parameter $\lambda$ which we sweep, reporting for each method the best result (best $\lambda$) in MC1 Accuracy that incurs at least equal performance in MMLU accuracy compared to the best (in terms of MC1 accuracy) of the two ACT methods (see L.2.2, giving 0.1% slack).

|  | Layer | Best $\lambda$ | MC1 Accuracy (%) ↑ | MC2 Accuracy (%) ↑ | MMLU Accuracy |
|---|---|---|---|---|---|
| Original | - | - | 25.46 | 40.27 | 65.35 |
| AURA | MLP | - | $25.34 \pm 0.15$ | $40.47 \pm 0.20$ | $65.37 \pm 0.06$ |
| ACTADD | Attention | 0.7 | $26.19 \pm 0.00$ | $40.88 \pm 0.00$ | $65.42 \pm 0.00$ |
| ITI-C | Attention | 1.0 | $27.42 \pm 0.30$ | $42.01 \pm 0.42$ | $65.26 \pm 0.11$ |
| Mean-ACT | Attention | 1.0 | $26.73 \pm 0.19$ | $42.20 \pm 0.24$ | $65.37 \pm 0.06$ |
| Linear-ACT | Attention | 1.0 | $27.17 \pm 0.23$ | $42.15 \pm 0.31$ | $65.33 \pm 0.11$ |
| ACTADD | All-LN | 1.0 | $25.58 \pm 0.00$ | $41.00 \pm 0.00$ | $64.88 \pm 0.00$ |
| ITI-C | All-LN | 3.0 | $29.65 \pm 0.71$ | $44.43 \pm 0.56$ | $64.71 \pm 0.22$ |
| Mean-ACT | All-LN | 1.0 | $32.88 \pm 0.54$ | $48.23 \pm 0.64$ | $64.83 \pm 0.14$ |
| Linear-ACT | All-LN | 1.0 | $33.22 \pm 0.22$ | $48.69 \pm 0.34$ | $64.78 \pm 0.15$ |
| ACTADD | MLP | 0.5 | $25.46 \pm 0.00$ | $40.64 \pm 0.00$ | $65.34 \pm 0.00$ |
| ITI-C | MLP | 2.0 | $30.11 \pm 0.60$ | $45.41 \pm 0.24$ | $64.71 \pm 0.14$ |
| Mean-ACT | MLP | 1.0 | $26.17 \pm 0.24$ | $41.27 \pm 0.34$ | $65.01 \pm 0.20$ |
| Linear-ACT | MLP | 1.0 | $26.41 \pm 0.52$ | $39.34 \pm 0.54$ | $60.98 \pm 3.14$ |

### L.2.2 SWEEPING $\lambda$ FOR ITI-C AND ACTADD

In Figures 13 - 16, we show the results of sweeping the value of $\lambda$ for ITI-C and ACTADD for both Gemma2-2B and Llama3-8B. For each model, we also indicate the MMLU accuracy of the best ACT method for that model with a horizontal grey dashed line, as this is our point of reference for choosing $\lambda$ for ITI-C and ACTADD: we choose the value of $\lambda$ that achieves the best MC1 accuracy, while achieving at least equal MMLU accuracy to this grey dotted line (up to a slack of 0.1%).

For ITI-C, where we see a clear relationship between MMLU and MC1 accuracy as $\lambda$ varies, we sweep $\lambda \in [1.0, 2.0, 3.0, 4.0, 5.0, 6.0, 7.0, 8.0, 9.0, 10.0, 11.0, 12.0, 13.0, 14.0, 15.0]$. For ACTADD,

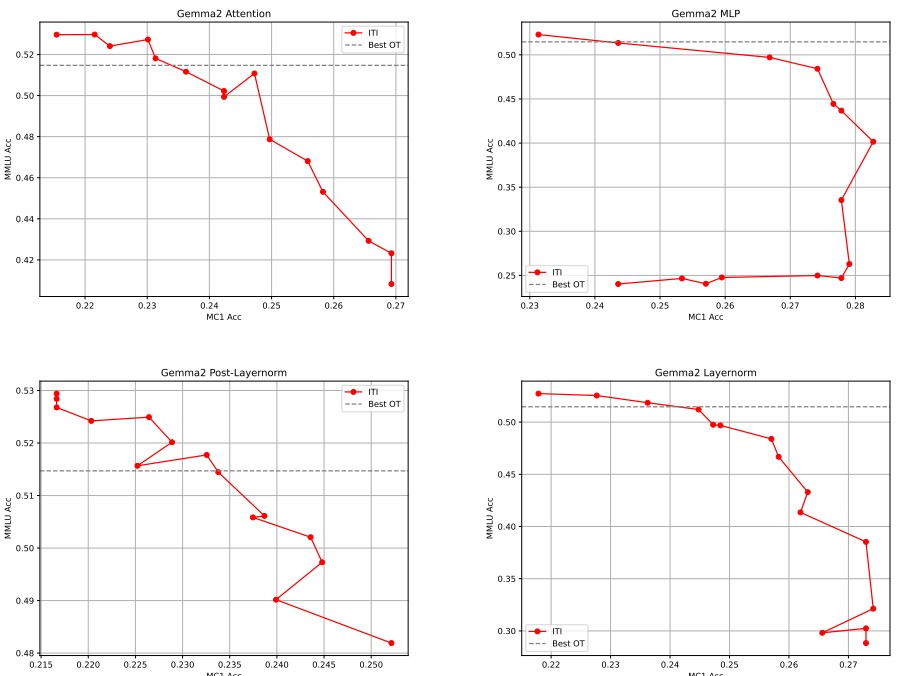

Figure 13: Sweeping $\lambda$ for inducing truthfulness with ITI-C on Gemma2-2B. Left endpoint of line is $\lambda = 1.0$, right endpoint of line is $\lambda = 15.0$ (each point increasing $\lambda$ by 1.0). Note this is for 1 seed only.

where the relationship can be more erratic, we also sweep values $< 1.0$. Here, we sweep $\lambda \in [0.1, 0.2, 0.3, 0.4, 0.5, 0.6, 0.7, 0.8, 0.9, 1.0, 2.0, 3.0, 4.0, 5.0]$.

Overall we see that $\lambda$ can have a strong impact on performance for ITI-C, but in a different way for each layer and model. In particular, it can decrease MMLU performance to catastrophic levels (more than halving performance on Gemma2-2B for MLP layers and on Llama3-8B for both attention and MLP layers), making it necessary to sweep $\lambda$ to find its value that provides a reliable control method using ITI-C for the problem at hand. Similar things can be found about ACTADD (e.g. when interventing upon on all Layernorm layers on Gemma2-2B, Figure 14).

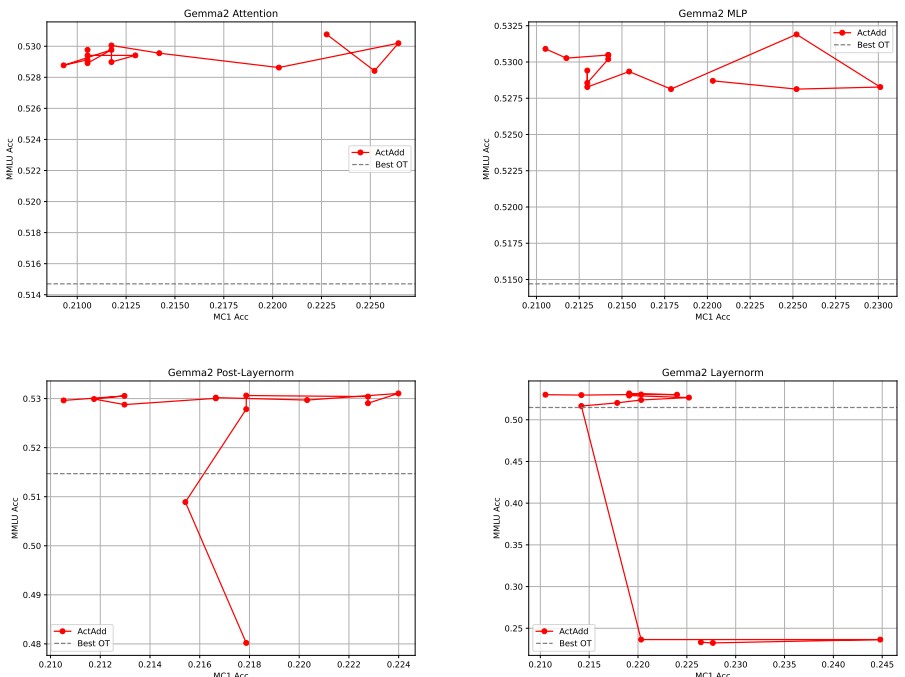

Figure 14: Sweeping $\lambda$ for inducing truthfulness with ACTADD on Gemma2-2B. Left endpoint of line is $\lambda = 0.1$, right endpoint of line is $\lambda = 5.0$ ($\lambda \in [0.1, 0.2, 0.3, 0.4, 0.5, 0.6, 0.7, 0.8, 0.9, 1.0, 2.0, 3.0, 4.0, 5.0]$). Note this is for 1 seed only.

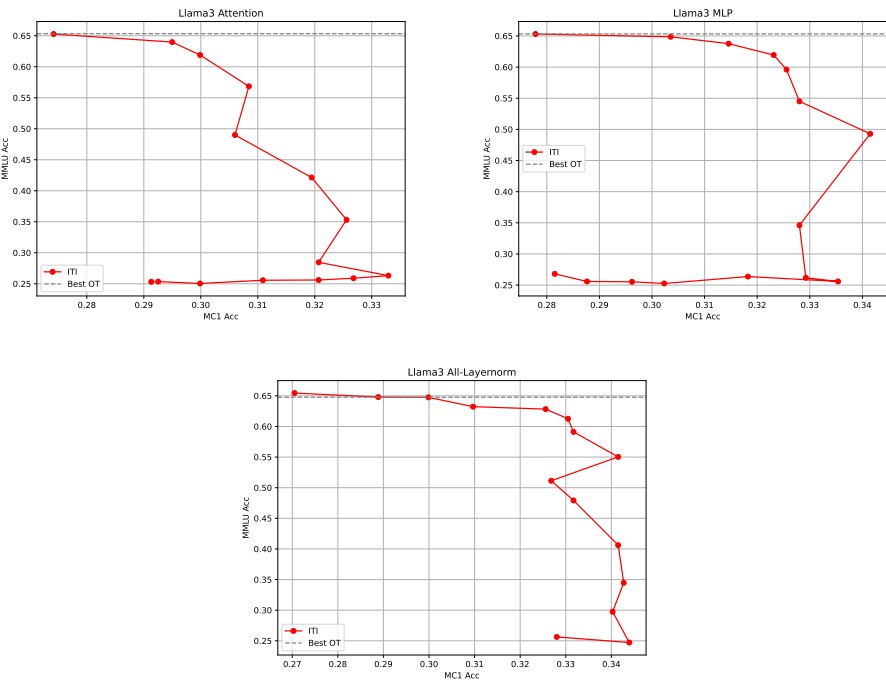

Figure 15: Sweeping $\lambda$ for inducing truthfulness with ITI-C on Llama3-8B. Left endpoint of line is $\lambda = 1.0$, right endpoint of line is $\lambda = 15.0$ (each point increasing $\lambda$ by 1.0). Note this is for 1 seed only.

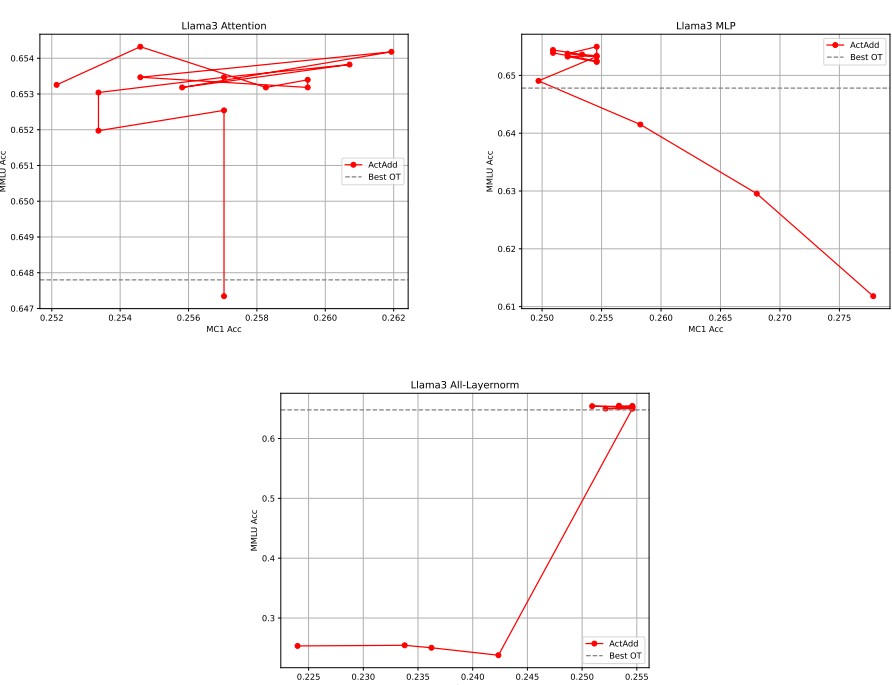

Figure 16: Sweeping $\lambda$ for inducing truthfulness with ACTADD on Llama3-8B. Left endpoint of line is $\lambda = 0.1$, right endpoint of line is $\lambda = 5.0$ ($\lambda \in [0.1, 0.2, 0.3, 0.4, 0.5, 0.6, 0.7, 0.8, 0.9, 1.0, 2.0, 3.0, 4.0, 5.0]$). Note this is for 1 seed only.

## M Experimental Details and Extended Results for T2I Generation

Appendix M.1 illustrates the effect of the guidance parameter in SDXL. Appendix M.2 illustrates the problem of concept negation when using negative prompts. Appendix M.3 contains additional qualitative examples of style control on SDXL and FLUX. Appendix M.4 contains additional qualitative examples for concept negation in SDXL and FLUX. Appendices M.6 and M.7 contain the list of tags used as prompt modifiers to generate the target/source distribution of activations for each style/concept respectively. Appendix M.8 contains details on FLUX's architecture conditioning.

### M.1 Guidance Parameter in Existing Diffusion Models

We show in Figure 17 the effect of changing the guidance scale parameter in SDXL. While large values lead to effective conditioning, lower values destroy content. This makes guidance non intuitive and harder to use by users.

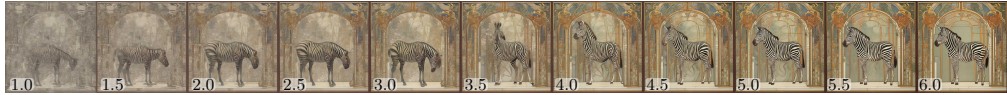

Figure 17: SDXL with *art nouveau* tags appended to the prompt as described in Appendix M.3 and guidance strength linearly increasing from 1 to 6. Note how for low guidance (left most images) the semantic content is almost completely lost.

### M.2 Negative Prompting

Stable diffusion models allow using negative prompts to avoid unwanted elements in the generated images (Rombach et al., 2022; Podell et al.). Here, we show that this method is ineffective at removing *pink elephant*, *white bear*, and *gorilla*. Figures 18 and 19 contain some failure cases of SDXL and Stable Diffusion 3 (Esser et al., 2024) at removing unwanted concepts. Figure 26 and Figure 27 show results intervening SDXL with AcT, showing its effectiveness at removing these concepts with the same prompts. In Figure 28 we show some failure cases at concept negation.

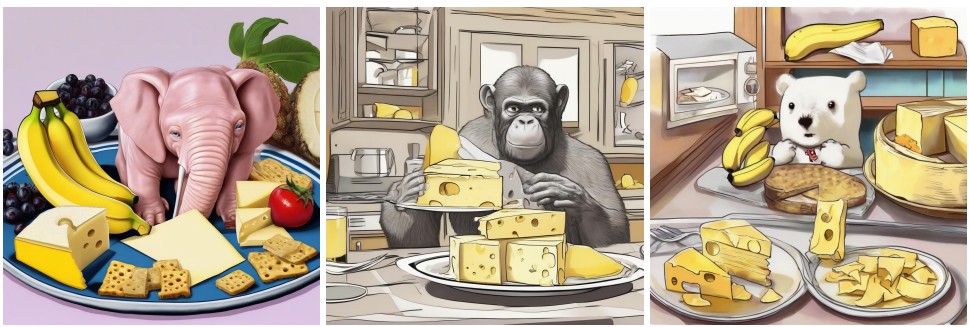

Figure 18: **SDXL with Negative Prompt.** Prompt: "There is a banana and two pieces of cheese on a plate. A {pink elephant, gorilla, white bear} cannot be seen anywhere.". Negative prompt: "A {pink elephant, gorilla, white bear}".

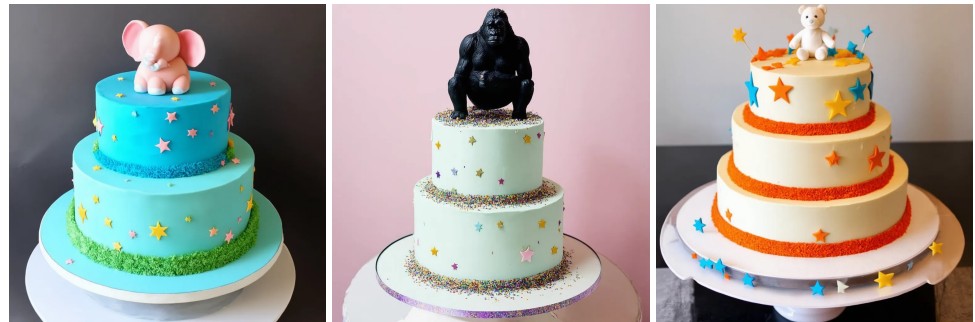

Figure 19: **Stable Diffusion 3 with Negative Prompt.** Prompt: "2 tier cake with multicolored stars attached to it. A {pink elephant, gorilla, white bear} cannot be seen anywhere." Negative prompt: "A {pink elephant, gorilla, white bear}.".

## M.3 STYLE CONTROL

Figures 20 to 22 complement the results shown in Section 5.1.

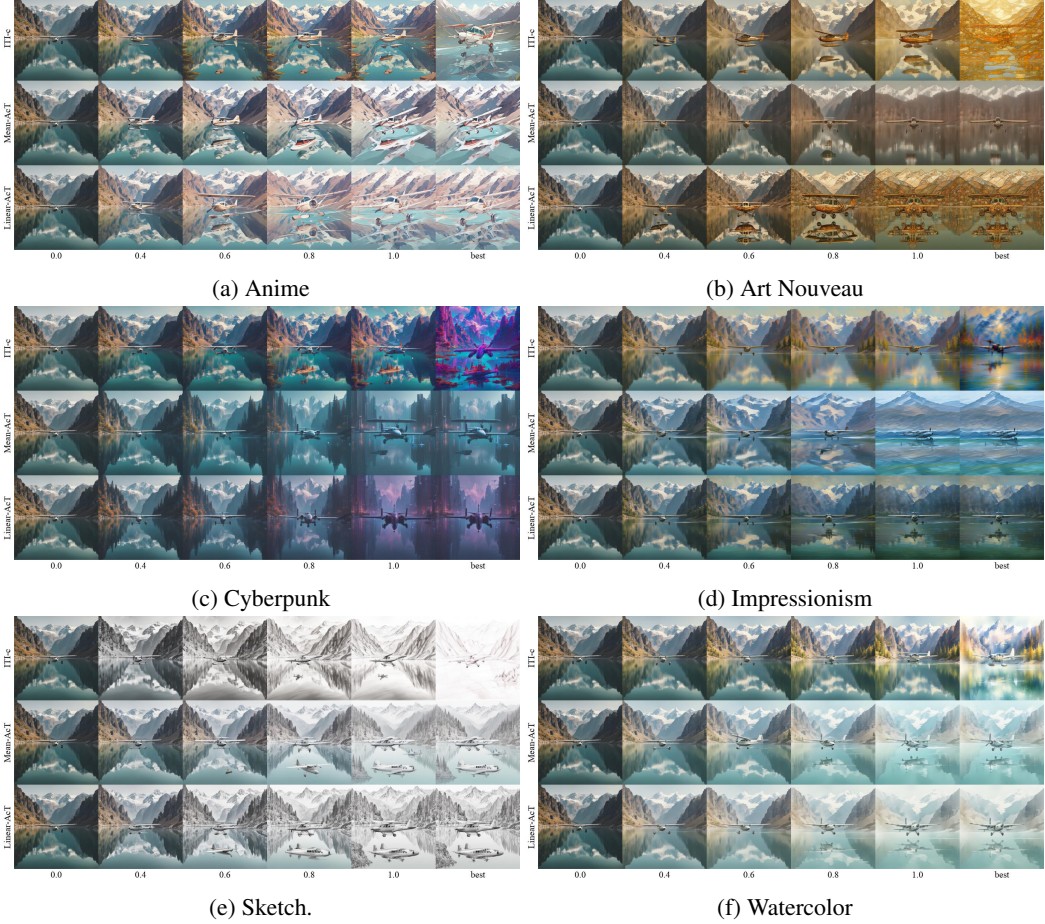

Figure 20: **SDXL - A plane floating on top of a lake surrounded by mountains.** From left to right conditioning strength $\lambda$ increases from 0 to 1. Rightmost column corresponds to the best strength found in Figure 6 ($\lambda = 1$ for AcT and $\lambda = 2$ for ITI-c). Linear-AcT succeeds at inducing different styles. Mean-AcT fails at inducing *art nouveau*. ITI-c introduces noise for *art nouveau* and *cyberpunk*.

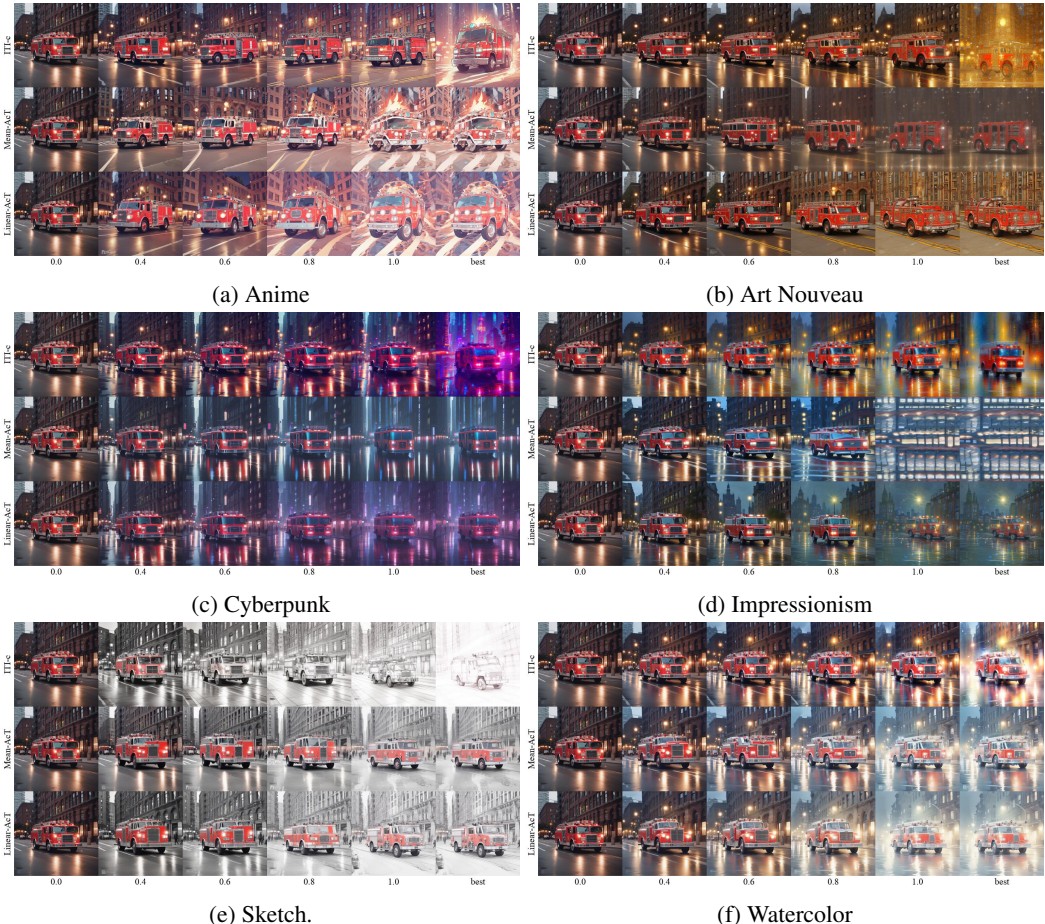

(a) Anime

(b) Art Nouveau

(c) Cyberpunk

(d) Impressionism

(e) Sketch.

(f) Watercolor

Figure 21: **SDXL - A firetruck with lights on is on a city street.** Rightmost column corresponds to the best strength found in Figure 6 ($\lambda = 1$ for ACT and $\lambda = 2$ for ITI-C). Mean-ACT fails at inducing *impressionism* and *art nouveau*. ITI-C achieves the strongest conditioning and generates a noisy image for *art nouveau*.

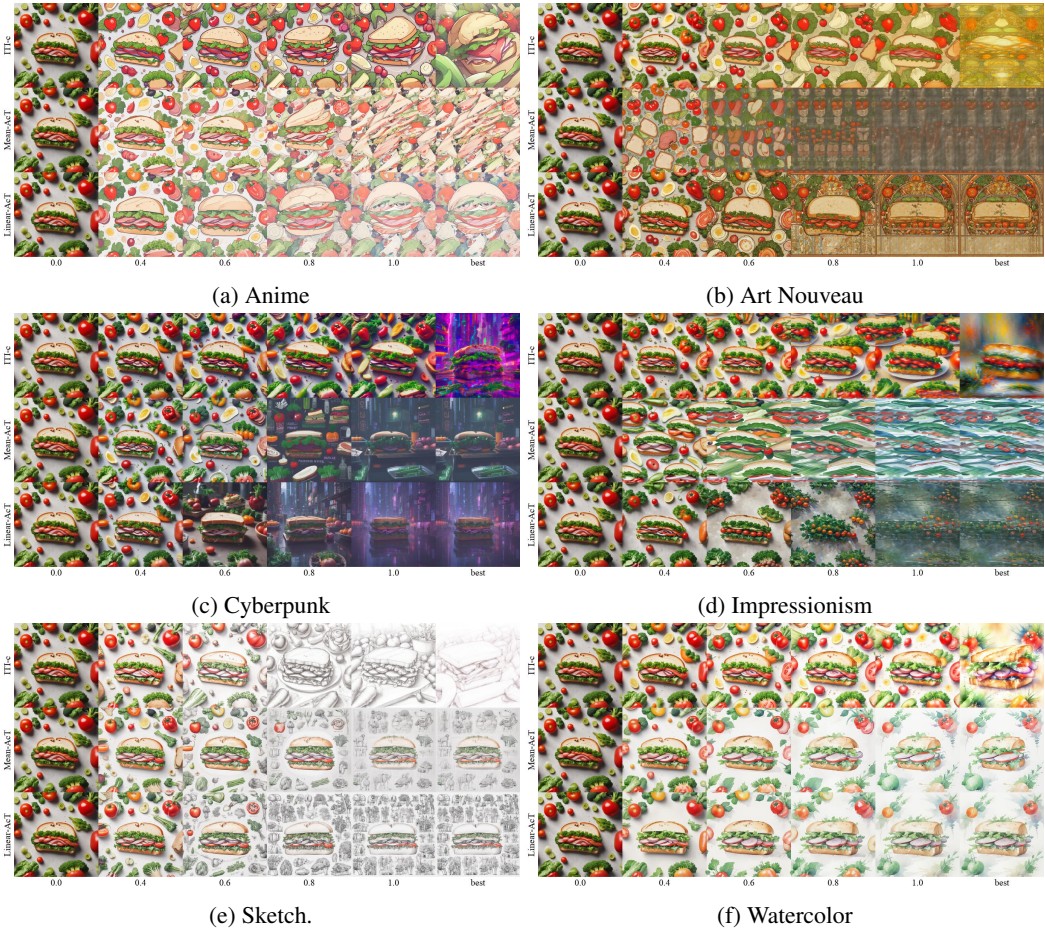

(a) Anime

(b) Art Nouveau

(c) Cyberpunk

(d) Impressionism

(e) Sketch.

(f) Watercolor

Figure 22: **SDXL - A sandwich is placed next to some vegetables.** Rightmost column corresponds to the best strength found in Figure 6 ($\lambda = 1$ for AcT and $\lambda = 2$ for ITI-c). ITI-c fails at inducing style progressively (e.g. (c) *cyberpunk*).

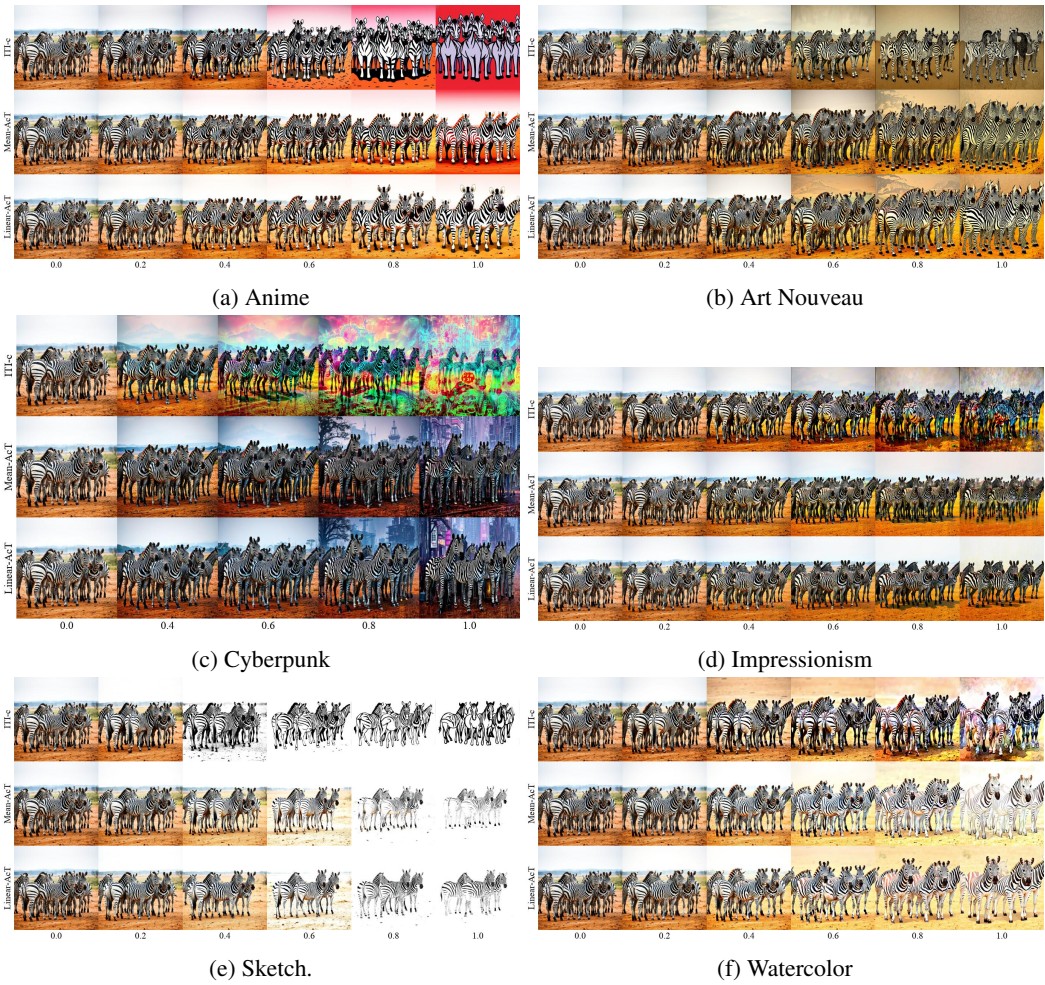

Figure 23: **FLUX - A group of zebra standing next to each other on a dirt field.** Rightmost column corresponds to the best strength found in Figure 6 ($\lambda = 1$ for all methods). Linear-AcT is successful at inducing all styles. ITI-c fails at inducing *cyberpunk* and *anime*.

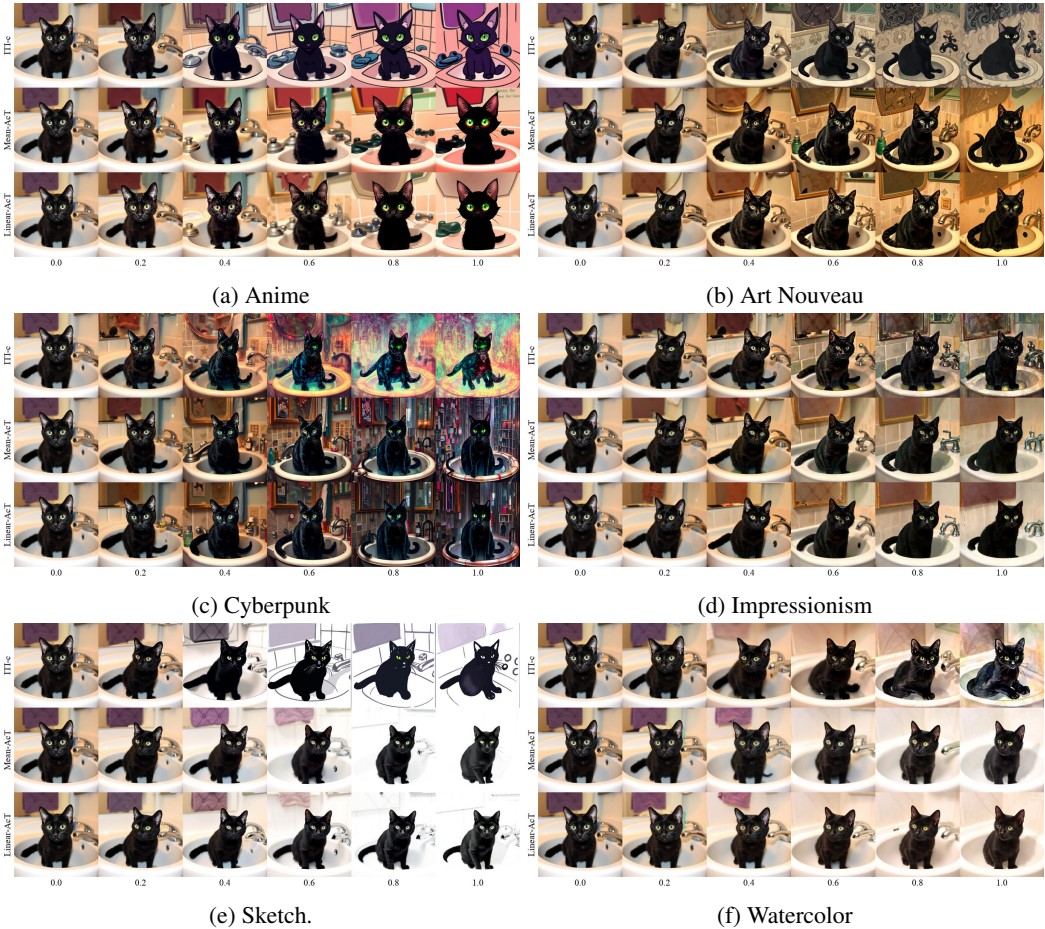

Figure 24: **FLUX - Black cat with green eyes sitting in a bathroom sink.** Rightmost column corresponds to the best strength found in Figure 6 ($\lambda = 1$ for all methods). AcT's conditioning is weak for *sketch* and *watercolor*. ITI-c fails at inducing *cyberpunk*.

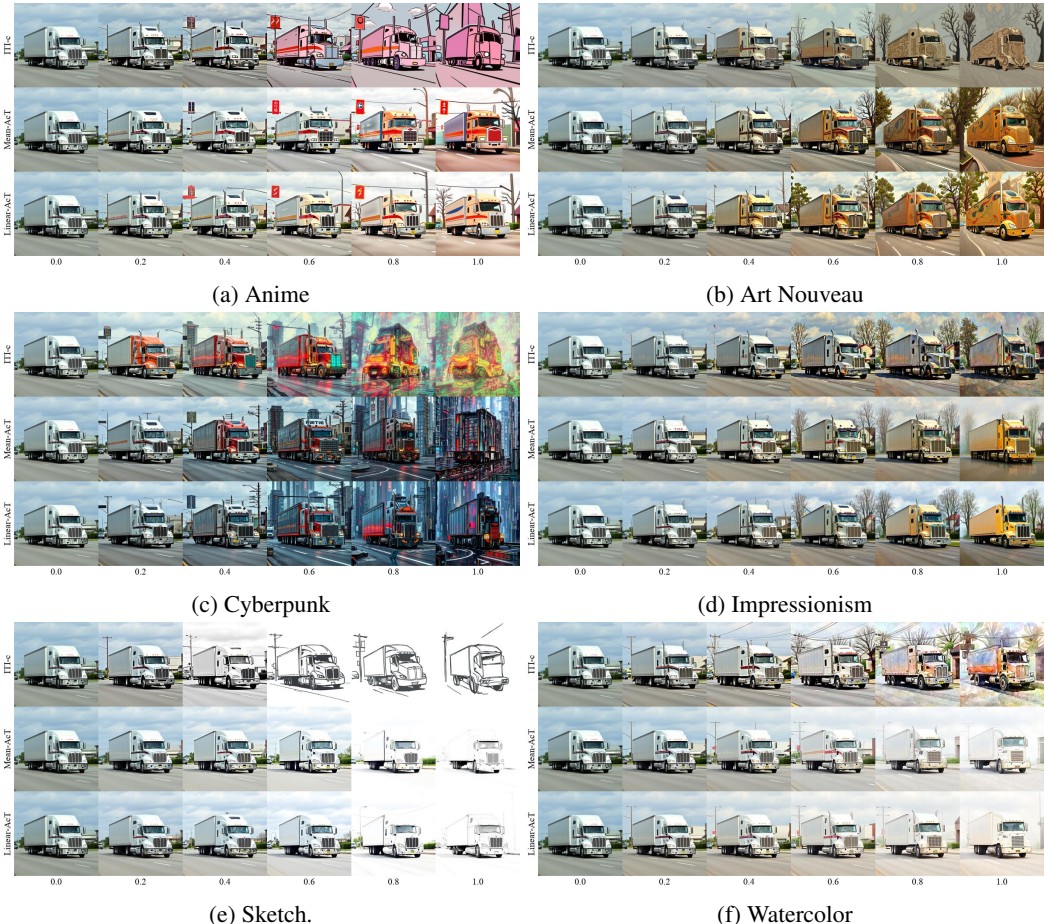

Figure 25: **FLUX - A semi truck is driving down a street.** Rightmost column corresponds to the best strength found in Figure 6 ($\lambda = 1$ for all methods). AcT is able to preserve the semantics for all styles and we observe only mild conditioning for *impressionism* and *watercolor*. ITI-C fails at inducing *anime* and *cyberpunk*.

## M.4 CONCEPT NEGATION

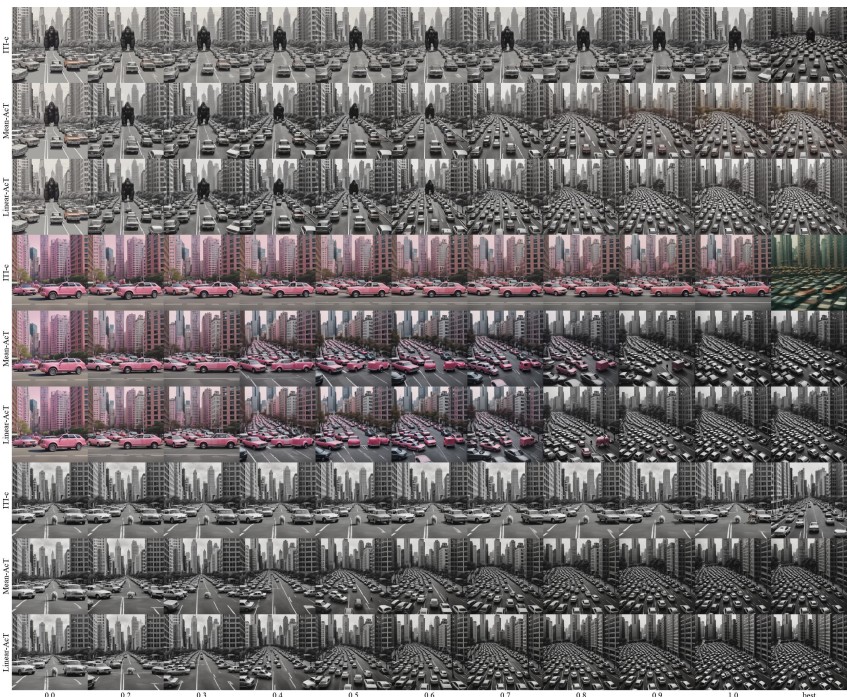

(a) Many cars parked on a city street with tall buildings in the background.

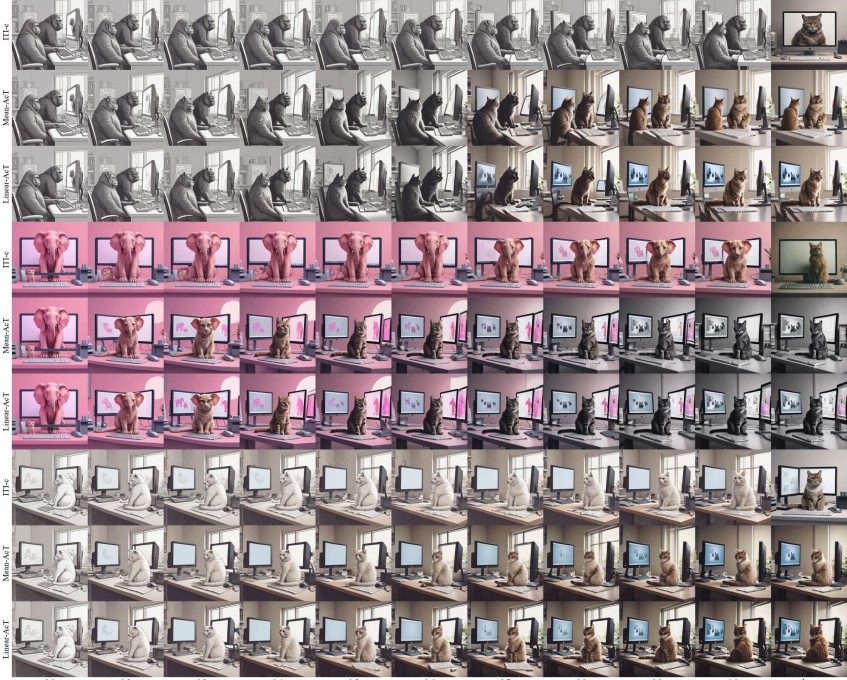

(b) A cat sitting in front of a large computer monitor.

Figure 26: **SDXL - Concept negation examples I.** Rightmost column corresponds to the best strength found in Figure 6 ($\lambda = 1$ for AcT and $\lambda = 4$ for ITI-c). Every 3 rows represent a different concept in {gorilla, pink elephant, white bear} which was negated at the input of the image generator. Mean-AcT and Linear-AcT succeed at removing the unwanted concept. ITI-c fails for *gorilla* and produces a blurry image for *pink elephant*.

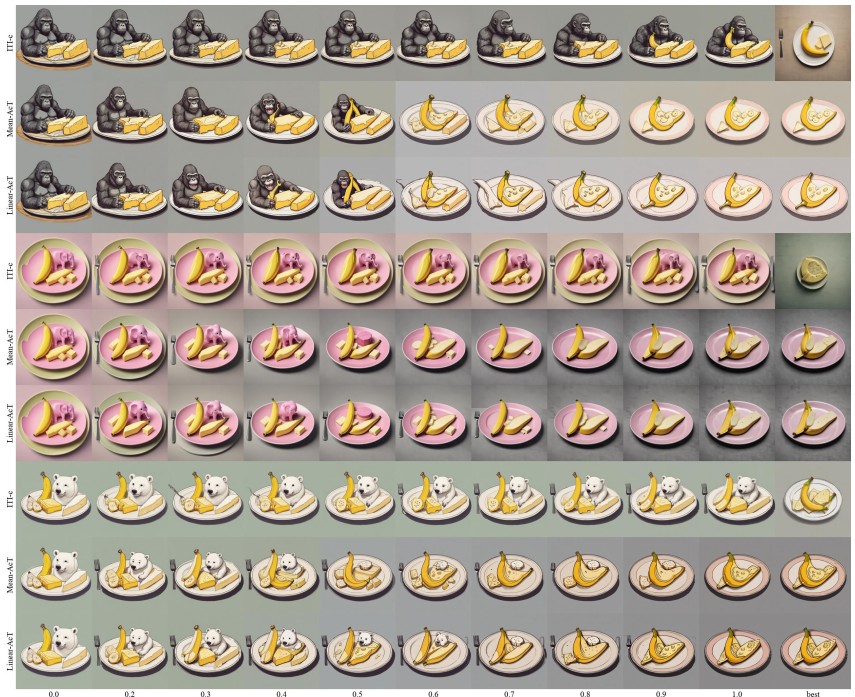

(a) There is a banana and two pieces of cheese on a plate.

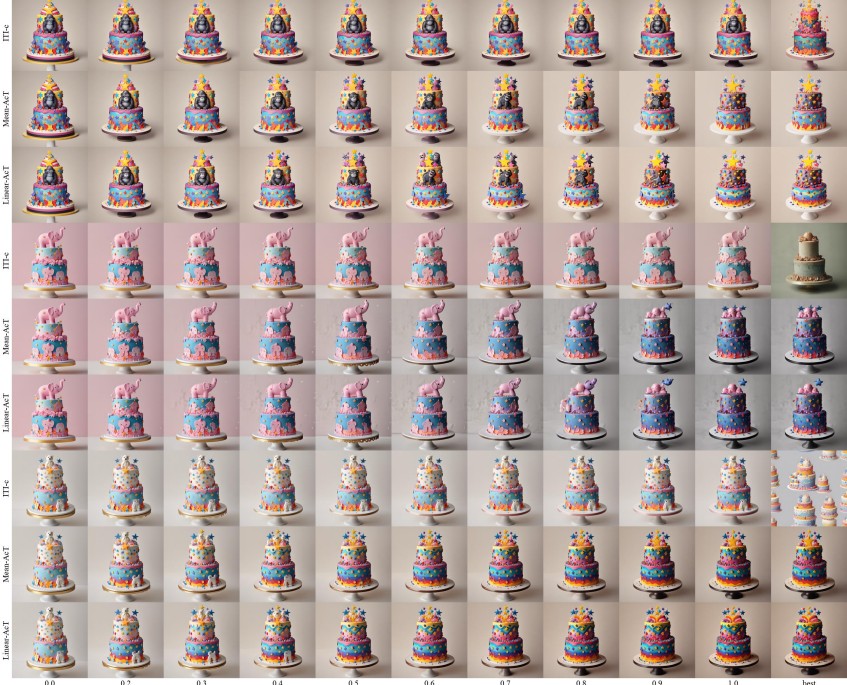

(b) 2 tier cake with multicolored stars attached to it.

Figure 27: **SDXL - Concept negation examples II.** Rightmost column corresponds to the best strength found in Figure 6 ($\lambda = 1$ for AcT and $\lambda = 4$ for ITI-c). Every 3 rows represent a different concept in {gorilla, pink elephant, white bear} which was negated at the input of the image generator. Linear-AcT and Mean-AcT succeed at removing the negated concepts while ITI-c tends to modify the semantics of the image.

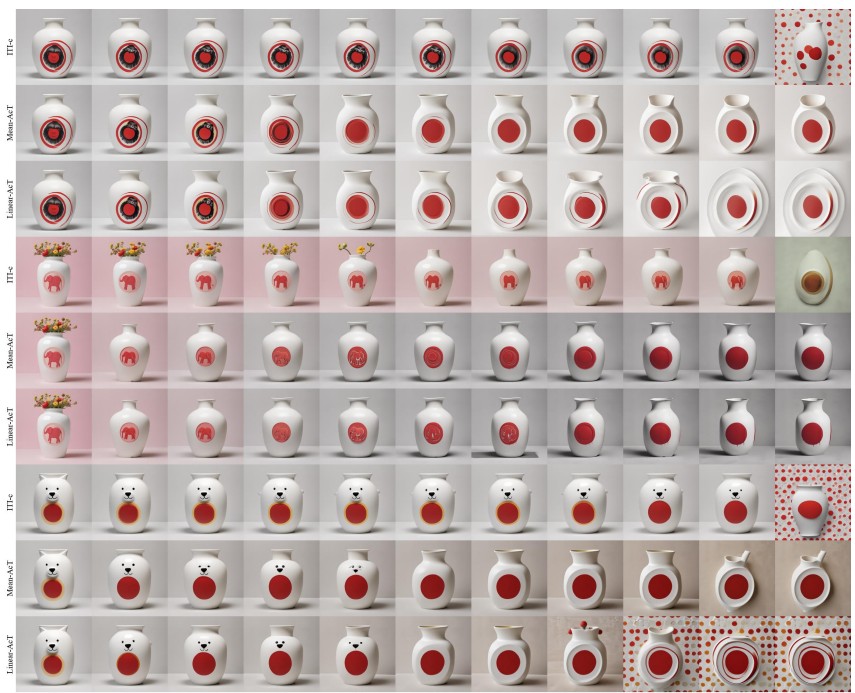

(a) Closeup of a white and yellow vase with a red circle at the bottom.

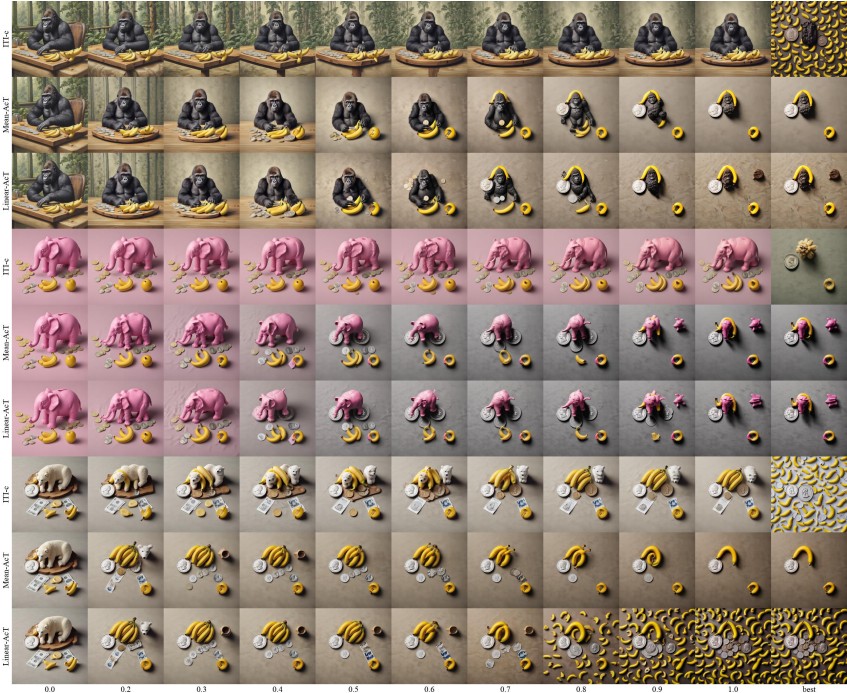

(b) A table topped with bananas next to a coin.

Figure 28: **SDXL - Concept negation examples III (failures)** Rightmost column corresponds to the best strength found in Figure 6 ($\lambda = 1$ for ACT and $\lambda = 4$ for ITI-C). Every 3 rows represent a different concept in {gorilla, pink elephant, white bear} which was negated at the input of the image generator. While Mean-ACT and Linear-ACT are successful at removing the concept, there is sometimes a change in semantics of the image for the maximum strength. ITI-C at best strength ($\lambda = 4$) changes semantics for all concepts.

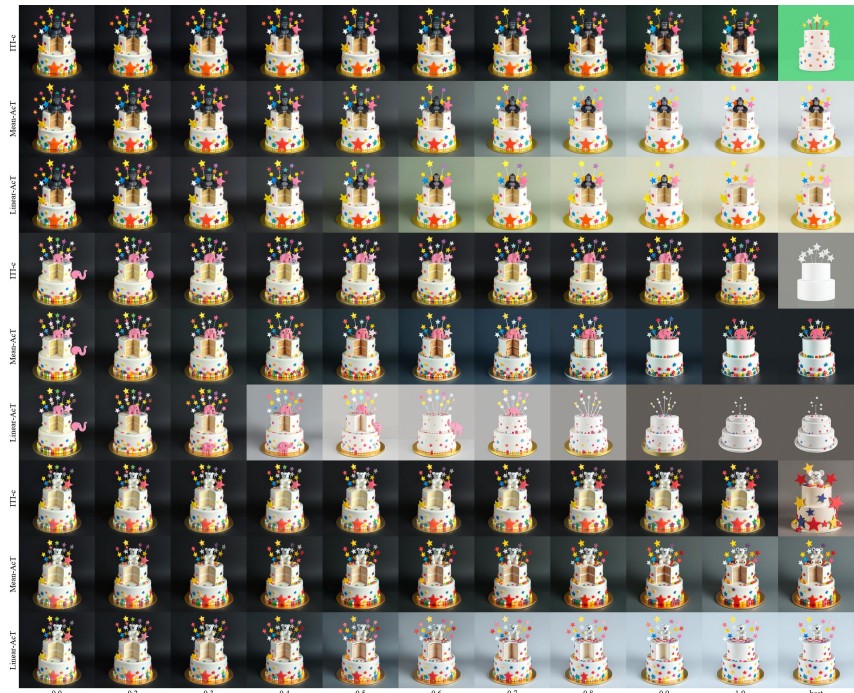

(a) 2 tier cake with multicolored stars attached to it.

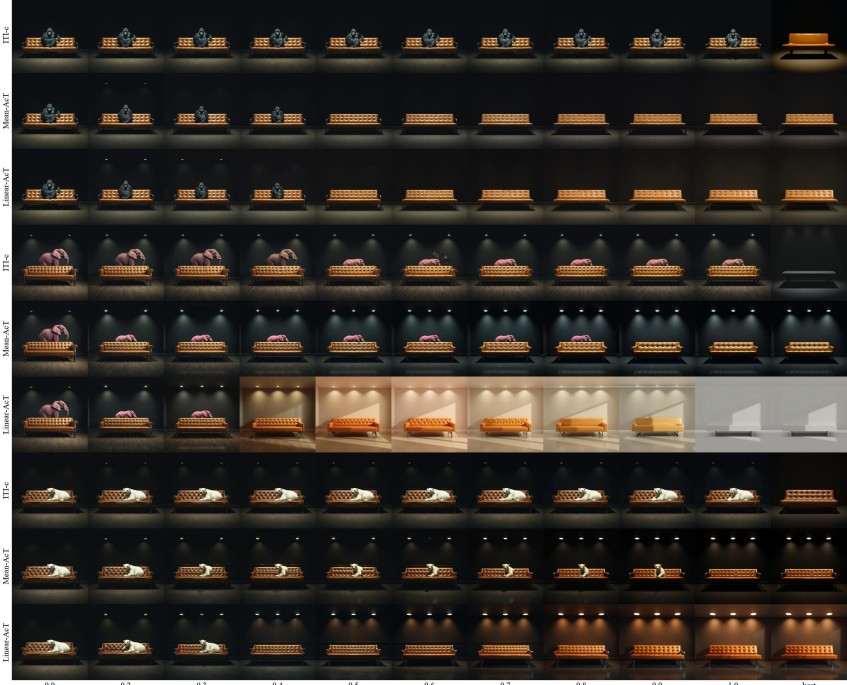

(b) A table topped with bananas next to a coin.

Figure 29: **FLUX - Concept negation examples I.** Rightmost column corresponds to the best strength found in Figure 6 ($\lambda = 1$ for AcT and $\lambda = 5$ for ITI-c). Every 3 rows represent a different concept in {gorilla, pink elephant, white bear} which was negated at the input of the image generator. Linear-AcT removes the negated concepts except for *white bear* in (a). ITI-c is effective at "best" ($\lambda = 5$). At high strengths, Linear-AcT and ITI-c also affect other image semantics.

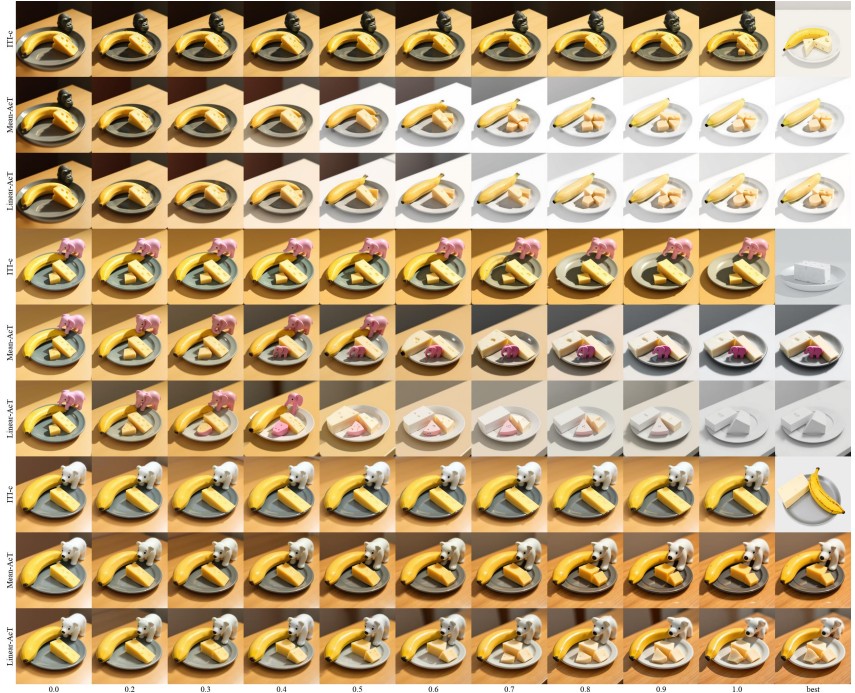

(a) There is a banana and two pieces of cheese on a plate.

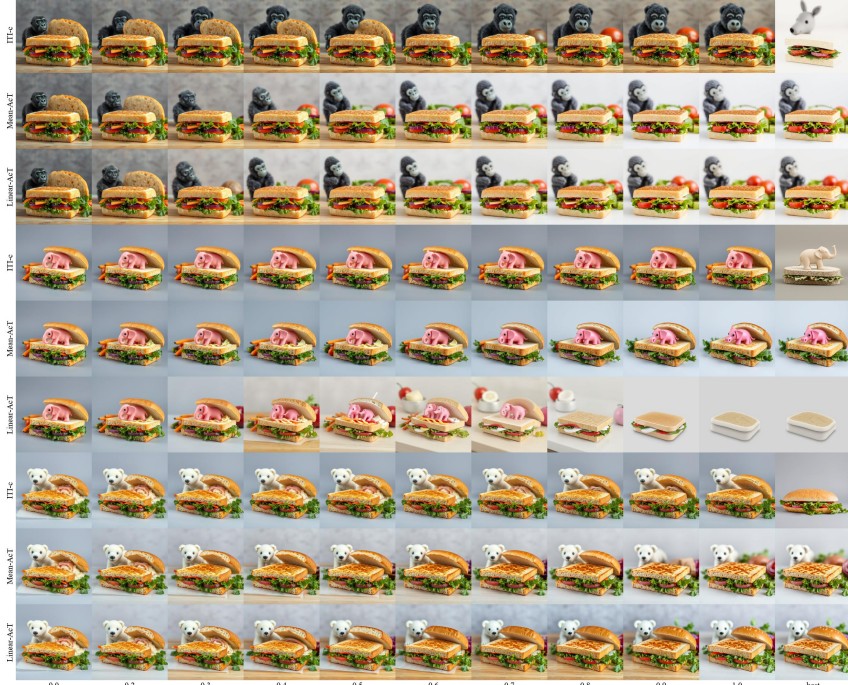

(b) A sandwich is placed next to some vegetables.

Figure 30: **FLUX - Concept negation examples II (Failures)** Rightmost column corresponds to the best strength found in Figure 6 ($\lambda = 1$ for AcT and $\lambda = 5$ for ITI-C). Every 3 rows represent a different concept in {gorilla, pink elephant, white bear} which was negated at the input of the image generator. AcT does not remove *white bear*, and fails to remove *gorilla* in (b). For high $\lambda$, Linear-AcT modifies the semantics of the image. ITI-C removes the unwanted concept for $\lambda = 5$.

## M.5 DETAILED RESULTS

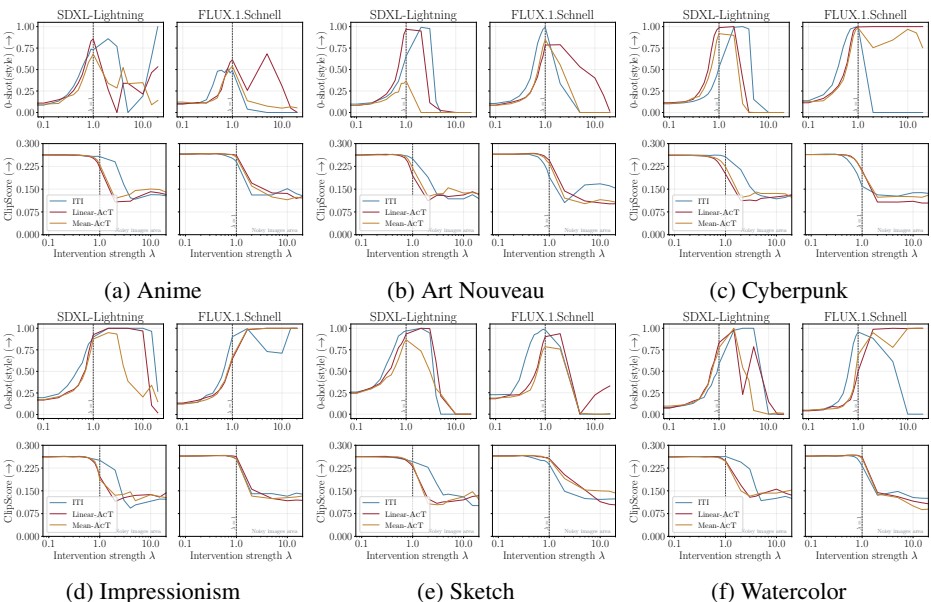

Figure 31: **Style induction.** For each style (a-f) and model (left-right), we show the 0-shot classification score for the style being present in the generated images (top) and the ClipScore to track how much generated images deviate from the unconditional prompt (bottom). The gray area indicates images that have lost their semantic content.

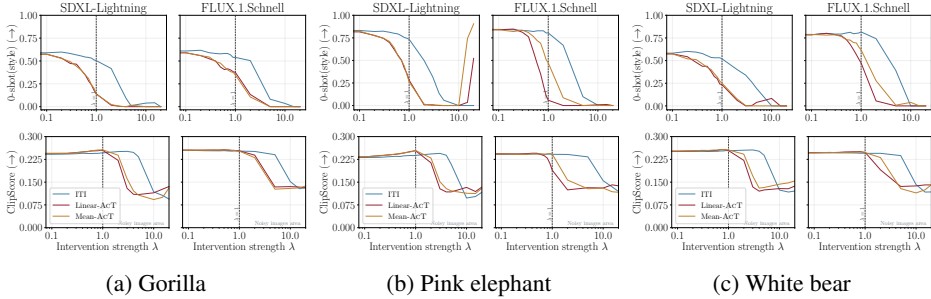

Figure 32: **Concept negation.** For each concept (a-c) and model (left-right), we show the 0-shot classification score for the concept being present in the generated images (top) and the ClipScore (bottom) to track how much generated images deviate from the unconditional prompt. The gray area indicates images that have lost their semantic content.

## M.6 STYLE PROMPTS

Table 15: List of tags generated with Llama-8B-instruct (right) to induce different styles (left).

| | |
|---|---|
| Anime | anime style, large expressive eyes, stylized hair, bold outlines, simplified colors, dynamic perspective, exaggerated features, angular shapes, chibis, manga inspired, emotive facial expressions, action sequences, speed lines, cell shading, graphic backgrounds, vibrant palettes |
| Art nouveau | Art Nouveau, Alphonse Mucha, Gustav Klimt, flowing lines, organic shapes, floral motifs, geometric patterns, ornamental designs, Jugendstil, Secessionism, symbolism, female figures, gold leaf, intricate details, turn of the century art, early 20th century |
| Impressionism | impressionism, Claude Monet, brush strokes, light, color, outdoor scenes, water lilies, haystacks, Rouen Cathedral, reflections, nature, atmospheric, vibrant colors, visible textures, 19th century art, French impressionism |
| Cyberpunk | cyberpunk, neon lights, urban jungles, high-tech architecture, augmented reality, AI technology, biopunk, futuristic cities, post-apocalyptic scenes, digital hacking, megacorporations, androids, dystopian societies, cybernetic enhancements, chromed details, glowing neon signs, rain-soaked streets |
| Photorealism | photorealism, hyperrealism, optical precision, photographic quality, fine detail, lifelike textures, realistic lighting, accurate perspective, human figures, still life, cityscapes, landscapes, skin tones, reflections and shadows, everyday objects, documentary style art, contemporary realism |
| Sketch | sketches, pencil drawing, charcoal sketches, ink illustrations, gestural lines, quick studies, figure drawing, perspective sketching, urban sketching, landscape sketches, still life drawings, sketchbook art, doodles, minimalist lines, expressive mark-making, observational drawing |
| Watercolor | watercolor style, transparent media, wet-on-wet application, dry brush strokes, soft blending, delicate touches, gentle shading, luminous hues, atmospheric lighting, ethereal quality, subtle textures, color gradients, painterly aesthetics, fluid paint behavior, watercolor paper texture |

## M.7 CONCEPT PROMPTS

Table 16: List of tags generated with Llama-8B-instruct (right) to induce different concepts (upper left) or to prompt models not to generate them (lower left).

| | |
|---|---|
| Pink elephant | `a pink elephant. containing a pink elephant. with a pink elephant in plain view. and a pink elephant. it displays a pink elephant. featuring a pink elephant. in addition to a pink elephant. and also a pink elephant. and a pink elephant as well. the pink elephant can be clearly seen.` |
| Gorilla | `a gorilla. containing a gorilla. with a gorilla in plain view. and a gorilla. it displays a gorilla. featuring a gorilla. in addition to a gorilla. and also a gorilla. and a gorilla as well. the gorilla can be clearly seen.` |
| White bear | `a white bear. containing a white bear. with a white bear in plain view. and a white bear. it displays a white bear. featuring a white bear. in addition to a white bear. and also a white bear. and a white bear as well. the white bear can be clearly seen.` |
| No pink elephant | `without a pink elephant. not containing a pink elephant. without a pink elephant in plain view. and a pink elephant that cannot be seen. it does not display a pink elephant. not featuring a pink elephant. lacking a pink elephant. and not a pink elephant. and a pink elephant is missing. the pink elephant cannot be seen.` |
| No gorilla | `without a gorilla. not containing a gorilla. without a gorilla in plain view. and a gorilla that cannot be seen. it does not display a gorilla. not featuring a gorilla. lacking a gorilla. and not a gorilla. and a gorilla is missing. the gorilla cannot be seen.` |
| No white bear | `without a white bear. not containing a white bear. without a white bear in plain view. and a white bear that cannot be seen. it does not display a white bear. not featuring a white bear. lacking a white bear. and not a white bear. and a white bear is missing. the white bear cannot be seen.` |

## M.8 DETAILS ON FLUX CONDITIONING

FLUX's diffusion architecture [4] is based on the transformer architecture (Vaswani et al., 2017). Concretely, it is composed of $N$ consecutive multi-modal fusion transformer residual blocks followed by $M$ uni-modal transformer residual blocks. We found that the most effective strategy for strong conditioning is to intervene upon the output of all blocks. However, we found that conditioning blocks closest to the output tends to deteriorate the generated images. Thus, we condition all the $N$ multi-modal blocks and the first 15 uni-modal blocks.

---

[4]https://blackforestlabs.ai/announcing-black-forest-labs/

