# OpenReview forum: "Controlling Language and Diffusion Models by Transporting Activations"
_ICLR.cc/2025/Conference — ICLR 2025 Spotlight_

### Official Review · Reviewer_eCSW · 2024-11-01

**Soundness:** 3
**Presentation:** 2
**Contribution:** 3
**Rating:** 8
**Confidence:** 3

**Summary:**

The paper presents a novel framework called Activation Transport (ACT) for controlling generative models by steering activations using optimal transport theory. It claims to offer fine-grained control over model behavior with minimal computational overhead. ACT is applied to tasks in both large language models (LLMs) and text-to-image diffusion models (T2Is), demonstrating effectiveness in mitigating toxicity, inducing concepts, and providing style control.

**Strengths:**

1. The use of optimal transport theory to steer activations is a novel idea that generalizes previous activation-steering methods.

2. ACT is shown to be effective across different types of models (LLMs and T2Is), which suggests broad applicability.

3. The method claims to add negligible computational cost, which is crucial for scalability in large models.

4. The paper provides experimental results demonstrating improvements in toxicity mitigation and style control, among other tasks.

**Weaknesses:**

1. The paper mentions that Linear-ACT assumes a linear transmission relationship between activations, which is a simplification made for computational and memory considerations. Is it possible for nonlinear mapping to allow more complex activation relationships and finer control?

2. Since the mapping estimate depends entirely on the samples used, its expressive power is limited by the diversity and coverage of the samples. How to improve the generalization ability of the mapping through more extensive data sampling or enhancement techniques.

3. The paper mainly focuses on the control of a single task, such as toxicity mitigation or style control. How to deal with multiple conflicting objectives or constraints simultaneously under a unified framework?

4. Although Linear-ACT has a certain robustness to the parameter λ, other methods such as ITI-C are very sensitive to the choice of model and layer, which may lead to a lot of parameter tuning in different situations.

5. Although the ACT method claims to have low computational overhead, there is a lack of detailed analysis of the specific computational costs and resource requirements of models of different sizes in practical applications.

**Questions:**

1. How does ACT perform in scenarios with highly multimodal distributions, where linear transport might be insufficient?

2. Can the authors provide more details on the computational costs associated with the implementation of ACT?

3. How sensitive is the method to the choice of the transport support $Q_o$ or $Q_∞$?

4. Are there any limitations observed in the diversity or creativity of outputs when using ACT for concept induction in LLMs?

5. How does the method handle potential biases introduced during the transport of activations?

---

> ### Author Response · Authors · 2024-11-20
> **Rebuttal to Reviewer eCSW [1/3]**
>
> We would like to thank you for this insightful review. We believe that the provided questions and suggestions have helped improve our manuscript. Find hereafter detailed answers to your comments and questions.
>
> > **W1. The paper mentions that Linear-ACT assumes a linear transmission relationship between activations, which is a simplification made for computational and memory considerations. Is it possible for nonlinear mapping to allow more complex activation relationships and finer control?**
>
> Thanks for this important comment. We did spend quite some time investigating other more advanced approaches (sigmoid-based non-linear maps, more complex linear maps with cross-variable effects, as well as [1] and [2]), and for now we discarded them as they yielded worse results due to optimization convergence and overfitting to the limited input data;  and required more hyper parameter tuning. Having robust maps becomes important when estimating them using the causal algorithm (Def 3.2), since mapping errors will compound throughout layers and affect the estimation of upstream maps. Taken together, this is why we settled on the linear choice, which showed to be robust, easy to interpret and incurred small costs in terms of memory and compute.
>
> We have added a short discussion on this within the current space limits (L.139 in revision, highlighted in blue) and will expand it in the appendix.
>
> That said, we agree with the reviewer that investigating more sophisticated maps while overcoming the challenges listed provides a very interesting follow-up research opportunity.
>
> *[1] Entropic estimation of optimal transport maps. Aram-Alexandre Pooladian, Jonathan Niles-Weed, arXiv:2109.12004*
>
> *[2] On amortizing convex conjugates for optimal transport. Brandon Amos. arXiv:2210.12153*
>
>
> > **W2. Since the mapping estimate depends entirely on the samples used, its expressive power is limited by the diversity and coverage of the samples. How to improve the generalization ability of the mapping through more extensive data sampling or enhancement techniques.**
>
> Thanks for raising this excellent point, we agree with you. For any data-driven method (like ours) data quality is of key importance. We believe this question is closely tied to the choice of OT estimation. We agree that data quality is critical, and we believe that variability in data should be handled in the map estimation pipeline itself.
> One possible avenue to further robustify the estimation is to change the cost in L.185. We have used the squared-Euclidean distance on the real line, but many other costs $c$ would work, e.g. absolute deviation or $p$-norm. This would require running the 2D minimization in L. 184 per neuron, rather than getting closed forms as in L.187, but would help *select* data samples during the map estimation. We are exploring this choice. We are also considering using unbalanced transport techniques. This opens a fairly wide hyper parameter space that would require significant experimentation.
>
>
> > **W3. The paper mainly focuses on the control of a single task, such as toxicity mitigation or style control. How to deal with multiple conflicting objectives or constraints simultaneously under a unified framework?**
>
> This is a very interesting direction for future work. For now, we see three possible extensions:
>
>
> * The simplest is to gather data representative of the multiple interventions taken jointly (either their intersection, union, etc...) and use our pipeline.
> * A more elaborate one would be to consider multiple interventions (say $K$), and consider a simplex of them (multiple $\lambda_1, \lambda_2, \dots, \lambda_K$ summing to 1) that provide weights to interventions. We believe that an easy extension would be to sum OT maps accordingly (e.g. define $\omega = \sum_i \lambda_i \omega_i$ and accordingly for $\beta$). This would be reminiscent of Wasserstein barycenters [1] that combine multiple maps.
> * An even more elaborate extension would be one that learns a parameterised family of Linear-ACT maps, akin to the hyper-network perspective adopted in [2]. In that scenario, all known interventions are encoded as a feature vector, and paired with their respective Linear-ACT interventions. This creates a supervision task where the goal is to regress the Linear-ACT intervention vs. the feature descriptor.
>
> We believe that the OT formalism to transport activation provides a very rich set of tools that is amenable to such extensions, but we believe that exploring them is beyond the scope of a single paper.
>
> *[1] Barycenters in the Wasserstein space, Martial Agueh, Guillaume Carlier, 2010*
>
> *[2] Supervised Training of Conditional Monge Maps, Charlotte Bunne, Andreas Krause, Marco Cuturi, Neurips 22*

---

> ### Author Response · Authors · 2024-11-20
> **Rebuttal to Reviewer eCSW [2/3]**
>
> > **W4. Although Linear-ACT has a certain robustness to the parameter λ, other methods such as ITI-C are very sensitive to the choice of model and layer, which may lead to a lot of parameter tuning in different situations.**
>
> We certainly agree with this observation. Indeed, being robust to the parameter $\lambda$ is a strong contribution of our work. Given the increasing number of generative models available, as well as the increasing number of tasks, such robustness is of paramount importance to provide a consistent user experience across models and tasks, but is also increasingly harder to achieve. We will clarify this point in the final manuscript, emphasizing the importance of the robustness to $\lambda$, and how  Linear-AcT provides better robustness. In the paragraph in L310 (of the revised manuscript) we discuss how other methods are less robust to $\lambda$.
>
> > **W5. Although the ACT method claims to have low computational overhead, there is a lack of detailed analysis of the specific computational costs and resource requirements of models of different sizes in practical applications.** +
> **Q2. Can the authors provide more details on the computational costs associated with the implementation of ACT?**
>
> We agree with you in that the current section on computational aspects (Appendix A) lacks details about the cost of estimating the transport maps, so we have updated Appendix A with the details below. The computational cost of AcT can be divided in two main parts: estimation cost, and inference cost.
>
> **The estimation cost** is the cost related to extracting activations from a model and estimation a transport map on top. Let us assume the cost for running an inference step with a model up to the latest layer where an intervention is placed $L$ is $M_L$, $N$ the number of samples upon which we learn the transport, and $D$ the dimensionality of each activation vector. We will also assume batch size = 1.
>
> *   Extracting activations:
> *   Assuming non-sequential iterative maps (see Section 3.2 in the submission): the cost for extracting activations is $\mathcal{O}(N M_L)$.
> *   Assuming sequential iterative maps, we need two forward passes per layer: the first is used to learn a transport map, and the second to produce responses after applying the map. Since the cost of applying a map with fixed strength is 0, the cost of extracting activations with iterative maps is $\mathcal{O}(2NM_L)$.
> *   Learning a linear transport map involves sorting $NLD$ activations for the source and target distribution and computing the affine transport params analytically (see Definition 3.1). Assuming half of the $N$ samples belong to the source and the target distributions respectively, the cost is dominated by the sorting operation $\mathcal{O}\left(NLD \log(NLD)\right)$(assuming quicksort is used), which is also smaller than the cost of a forward pass through the model.
>
> **The inference** The inference cost is the cost related to generating an output with an intervened model. As explained at the beginning of the section, assuming a fixed transport map strength ($\lambda$), the affine transport map can be directly fused into the model weights and thus the additional cost of \linear is $O(0)$. If we need to be able to tune the intervention strength, then we cannot fuse it into the weights and the cost is that of a 1-d affine map on all the transported activations, which is significantly smaller than the cost of a forward pass on the model, which involves expensive matrix multiplication: $\mathcal{O}(LD)<<\mathcal{O}(M)$.
>
> In summary, estimation is only done once, has cost $\mathcal{O}(NM_L)$, and it is amortized during inference. The inference transport cost is $O(0)$ with fixed $\lambda$ (AcT can be fused into linear layers and **has no cost at inference time**) and $\mathcal{O}(LD)$ with variable $\lambda.$ In other words, estimating a transport map is much cheaper than training a model and has no impact at inference time unless one needs control over $\lambda$, in which case the additional cost is significantly smaller than the cost of a forward pass.
>
> > **Q1. How does ACT perform in scenarios with highly multimodal distributions, where linear transport might be insufficient?**
>
> While we do observe that most marginals are unimodal *by design* when training LLMs, this situation does not preclude, indeed, having multiple modes when looking at the distribution holistically across all dimensions (even if a small number of $n$ features are bimodal, this could generate potentially $2^n$ modes for the multidimensional distribution).
>
> That being said, in the current setup, for which the sample size of the data we use is roughly comparable to dimensionality, by the same argument (sample size $n\ll 2^k$ even for small $k$) , the budget of samples is not sufficient to detect multiple modes in high dimensions. Again, in this regime linear-ACT seems to be the most robust choice.

---

> ### Author Response · Authors · 2024-11-20
> **Rebuttal to Reviewer eCSW [3/3]**
>
> > **Q3. How sensitive is the method to the choice of the transport support $Q_0$  or $Q_\infty$ ?**
>
> The choice of transport support is specially important when transporting from *narrow* to *broad* distributions (eg. toxic to non-toxic), since it prevents transporting activations that are far from the domain observed during the estimation of the map. Such transports could be wrong (since no data was available at estimation) and harm the overall inference process.
> In appendix D of the submitted manuscript (Appendix E of the updated manuscript) titled *“The Effect of the Transport Support”* we provide experimental support of this intuition in the toxicity mitigation setup, showing that $Q_0$ offers a better trade-off between conditioning and utility degradation than $Q_\infty$ for that setup.
>
>
> > **Q4. Are there any limitations observed in the diversity or creativity of outputs when using ACT for concept induction in LLMs?**
>
> We believe this is an important question. To answer it, we measure the Self-BLEU score [1] for the sets of generated sentences after RTP prompts. Note that smaller Self-BLEU scores indicate higher diversity in a set of sentences, while large Self-BLEU shows repeatedness in the sentences. For example, a set of identical sentences will return a Self-BLEU of 1.
>
> The Self-BLEU results are summarized in the table below. We observe that Linear-AcT ($\text{Self-BLEU}=0.134$) better preserves the diversity shown by the non-intervened model ($\text{Self-BLEU}=0.130$). In this setting, ITI-c achieves 0.144 and our Mean-AcT a Self-BLEU of 0.140. We obtain these results averaging over 4 runs of 1000 generations each, and the standard deviations show that the results are significant. These experiments use same layer and $\lambda$ configuration reported in Table 2 for ITI-c (attention, $\lambda=8$) and for Mean-AcT/Linear-AcT (post-LN, $\lambda=1$). The provided results and discussion have been added to Appendix G of the updated manuscript.
>
> |Method    |Self-BLEU $(\downarrow)$    |
> |---    |---    |
> |None    |$0.130 \pm 0.003$    |
> |ITI-c      |$0.144 \pm 0.003$    |
> |Mean-AcT    |$0.140 \pm 0.002$    |
> |Linear-AcT    |$0.134 \pm 0.002$    |
>
> Besides the quantitative experiment, one intuition we would like to share is that Linear-AcT maps activations in-distribution for any $\lambda\in [0, 1]$. We hypothesize that this property of Linear-AcT helps preserve diversity, effectively avoiding collapse situations during inference due to OOD activations.
>
> *[1] Zhu, Yaoming, et al. "Texygen: A benchmarking platform for text generation models." The 41st international ACM SIGIR conference on research & development in information retrieval. 2018.*
>
>
> > **Q5. How does the method handle potential biases introduced during the transport of activations?**
>
> Thank you for raising this important point. As other data-driven approaches, our method will potentially reflect biases in the data used. For example, without specific data curation, transporting from man to woman could also move occupations from doctor to nurse [1]. However, our method has some properties that can make bias mitigation easier. First, we need small amounts of data (tenths or hundreds of sentences) to estimate Linear-AcT, which makes curation and human inspection easier. Second, our method is interpretable given the linear and independent nature of the maps learnt. This allows to inspect what each map is doing at neuron level and implement further bias-related guardrails based on map behavior.
>
> *[1] Kotek, Hadas, Rikker Dockum, and David Sun. "Gender bias and stereotypes in large language models." Proceedings of the ACM collective intelligence conference. 2023.*

---

> > ### Comment · Reviewer_eCSW · 2024-11-26
> >
> > I appreciate the author's comprehensive response, which addressed most of my concerns. I think this is a good paper, so I raised the score to 8.
> >
> > However, since I am not an expert in this field, my comments are for reference only.

---

> > > ### Author Response · Authors · 2024-11-26
> > > **Answer to Reviewer eCSW**
> > >
> > > We would like to thank you for your time reading our rebuttal, as well as for your comments. We are very happy to hear that our answers have led you to increasing your score. We remain available until the end of the rebuttal period to answer any other questions you may have.

---

### Official Review · Reviewer_WjWp · 2024-11-03

**Soundness:** 3
**Presentation:** 4
**Contribution:** 3
**Rating:** 8
**Confidence:** 4

**Summary:**

This paper proposes Activation Transport (AcT) as a general approach to steer activations in generative models, to map activations from one source distribution to the target distribution, while minimizing limitations of current inference intervention approaches. The authors show that the optimal transport-based framework proposed here generalizes many of the previous activation steering work. More specifically, AcT accounts for difference in the amount of variance for each activation dimension in its map which previous approaches do not. The representation maps are iteratively derived updated layer by layer, starting from the earlier layers to the last layers. AcT is agnostic to the modalities; as shown in the experiments, AcT can steer generation in the text domain, in applications such as toxicity mitigation, concept induction and truthfulness, and in the image generation, in applications such as style control and concept negation.

**Strengths:**

The proposed approach is grounded, intuitive and is applicable to several modalities and domains in generative modeling.

The paper is well written and easy to follow.

Extensive experiments are conducted to compare AcT to baselines approaches in text and image generation.

**Weaknesses:**

Scope is limited to single modality within a model and linear (not non-linear) mapping.

Experiments and results supporting the claim of AcT better preventing representations from being OOD is lacking.

**Questions:**

Choice of pooling operation: Can the authors show data to support the choice of average pooling? Some discussion about why average pooling is ideal versus max pooling or last token would be insightful. Ablation disentangling the choice of pooling layers versus other approaches would better differentiate the contribution of AcT and the pooling operations.

---

> ### Author Response · Authors · 2024-11-20
> **Rebuttal to Reviewer WjWp**
>
> We would like to thank you for appreciating our work. The very interesting comments and questions you have provided in this review have led us to conduct new experiments that, we believe, improve our manuscript. Please find detailed answers hereafter.
>
> > **W1. Scope is limited to single modality within a model and linear (not non-linear) mapping.**
>
> Controlling multi-modal models is certainly an exciting research avenue. In this work, we have purposely tackled conditioning using text inputs. Our experiments with text-to-image Diffusion are already going in the direction of multi-modality, since it uses text as input but the transport is done on the diffusion UNet. Using multiple modalities as inputs would require a whole new exploration which is beyond the scope of this work, since one should account with interdependencies between modalities. For example, how complementary are sentence and images about the same concept? Which modalities are most effective for conditioning? How do we blend activations steered with different modalities? And many more exciting but unsolved research questions.
>
> Answering your comment on non-linear maps, we did spend quite some time investigating other more advanced approaches (sigmoid-based non-linear maps, more complex linear maps with cross-variable effects, as well as [1] and [2]), for now we discarded them as they yielded worse results (due to optimization convergence and overfitting to the limited input data) and required more hyper parameter tuning. Taken together, this is why we settled on the linear choice, which showed to be robust and easy to interpret. We have added a comment in that direction (L.139 in the updated manuscript, all new changes appear in blue).
>
> That said, we agree with the reviewer that investigating more sophisticated maps while overcoming the challenges listed provides a very interesting follow-up research opportunity.
>
> *[1] Entropic estimation of optimal transport maps. Aram-Alexandre Pooladian, Jonathan Niles-Weed, arXiv:2109.12004*
>
> *[2] On amortizing convex conjugates for optimal transport. Brandon Amos. arXiv:2210.12153*
>
> > **W2. Experiments and results supporting the claim of AcT better preventing representations from being OOD is lacking.**
>
> This is a very good point, and we agree that showing evidence of the effect of different interventions on the internal distributions can strengthen the claims in our paper. Following your suggestion, we have included new results where we plot the distributions of different activations at different layers of Gemma2-2b before (toxic sentences) and after applying Linear-AcT and ITI-c (detoxified), and compare them to the *true* destination distribution (non-toxic sentences). We choose ITI-c for this experiment as a representative of vector-based interventions that yields good results in many tasks. To better illustrate the effect of the intervention, for each layer, we choose those activations with larger distribution separation (measured with a normalized version of the optimal transport cost). The results show that Linear-AcT preserves the destination distribution while ITI-c does not in many cases. We use the optimal strength for each method, as found in the toxicity experiments (i.e. $\lambda=1$ for Linear-AcT and $\lambda=8$ for ITI-c).
>
> These new results and discussion have been added in Appendix F in the updated manuscript.
>
>
> > **Q1. Choice of pooling operation: Can the authors show data to support the choice of average pooling? Some discussion about why average pooling is ideal versus max pooling or last token would be insightful. Ablation disentangling the choice of pooling layers versus other approaches would better differentiate the contribution of AcT and the pooling operations.**
>
> Thank for suggesting this! We have ran a new ablation over the following pooling operations for the toxicity setup on Gemma-2-2b:
>
>
> ```
> # Pooling operations considered,
> # assuming x to be a tensor of dimensions (batch x seq_len x dim)
> min = lambda x : x.min(1)
> max = lambda x : x.max(1)
> mean = lambda x: x.mean(1)
> last = lambda x: x[:, -1, :]
> ```
>
>
> The results in the ablation table below confirm that mean pooling achieves a better trade-off between toxicity reduction and model degradation (measured through MMLU) than other pooling operations. We have included this ablation in Appendix D in the revised manuscript.
>
> |Method    |Pooling $\phi$    |Strength $\lambda$    |CLS Tox. ($\downarrow$)    |MMLU ($\uparrow$)    |
> |---    |---    |---    |---    |---    |
> |Original    |None    |    |$4.17 \pm 0.32$    |$53.06$    |
> |Linear-AcT    |max    |1    |$1.80 \pm 0.12$    |$47.01 \pm 0.30$    |
> |Linear-AcT    |min    |1    |$0.77 \pm 0.12$    |$45.85 \pm 0.09$    |
> |Linear-AcT    |last    |1    |$0.47 \pm 0.17$    |$48.49 \pm 0.25$    |
> |Linear-AcT    |mean    |1    |$0.70 \pm 0.10$    |$51.87 \pm 0.06$    |

---

### Official Review · Reviewer_CvQ7 · 2024-11-04

**Soundness:** 3
**Presentation:** 3
**Contribution:** 3
**Rating:** 6
**Confidence:** 4

**Summary:**

This paper studies the problem of activation steering for diffusion models and Transformer language models. The key idea is to view activation steering as optimal transport. Under this umbrella, most existing methods are equivalent to mean transport map, which might not be optimal and do not preserve the activation distribution. Instead, the proposed method, Linear-ACT, can avoid out-of-distribution activations after steering.

Experiments on Transformer language models (Gemma and Llama) show that the proposed method outperform baselines that use constant vectors for activation steering, on tasks including toxicity mitigation, concept inducing, and truthfulness. Experiments on diffusion models (Stable Diffusion models) show the effectiveness of the proposed method for style control and concept negation.

**Strengths:**

- This paper studies an interesting problem in activation steering -- out-of-distribution activations. Many existing works require a very large coefficient before the steering vector, which can easily lead to OOD activations. The proposed method instead do not need this extrapolation.
- The experiments are extensive, covering a wide range of control tasks for language models and diffusion models. Many of them are important tasks such as truthfulness and style control.
- The paper proposes a unified framework to understand the connection between different activation steering methods.

**Weaknesses:**

- One of the exciting applications of activation steering or representation engineering is safety. It would be interesting to see how well the proposed method perform on safety risk mitigation.
- The baselines are mainly vector addition methods. I wonder how the proposed method compare with vector projection methods such as https://arxiv.org/abs/2303.02536

**Questions:**

N/A

---

> ### Author Response · Authors · 2024-11-20
> **Rebuttal to Reviewer CvQ7**
>
> We would like to thank the reviewer for many encouraging comments, and two thought-provoking questions that we have answered below. They have led to small edits in the revised paper that we invite the reviewer to check.
>
> > **W1. One of the exciting applications of activation steering or representation engineering is safety. It would be interesting to see how well the proposed method perform on safety risk mitigation.**
>
> We agree with you, safety it is definitely an exciting area where AcT could play a key role. We have shown some of these applications in the paper: we have provided results on reducing toxicity in LLMs (Sec 4.1), inducing truthfulness in LLMs (Sec 4.3), and for diffusion models we have shown the ability to prevent specific concepts from being generated in images (Sec 5.2). To avoid using images that could hurt the sensitivity of the reader we used playful concepts such as “pink elephant”, however, a straightforward application for safety would be, for example, preventing generation of images that contain nudity or other “not suitable for work” content.
>
> There are a number of other applications that could benefit from the blueprint of AcT interventions. We would be extremely interested in new benchmarks or suggestions on that topic that could form the basis of future work.
>
> > **W2. The baselines are mainly vector addition methods. I wonder how the proposed method compare with vector projection methods such as https://arxiv.org/abs/2303.02536**
>
> We thank the reviewer for pointing us to this interesting work, which we refer to now in the main body (Lines 59 & 104). While we understand its importance within the general framework of post-hoc actions on activations which definitely warrants a citation in our work, we are not sure how to include it in our work as a baseline, as you suggest.
>
> To our understanding, this work proposes a method (**Distributed Alignment Search or DAS**) that succeeds at aligning high-level causal models with low-level neural representations. DAS has the ability to learn interventions involving interactions between multiple neurons simultaneously (with increased memory footprint), as opposed to ACT's focus on single-neuron adjustments. This capacity can offer deeper insights into neural representations and causal structures in interpretability-focused scenarios.
>
> However, it is worth noting that the experiments in the DAS paper primarily focus on interpretability tasks. While the method could theoretically be used to induce specific behaviors in generative models, as it stands, the DAS paper does not propose or evaluate a method for controlling generations or intervening at inference time.
>
> Following your comment, we have tried to imagine what such an intervention would look like, but have struggled with this and reached a possible adaptation. For the example of toxicity, would the idea be to learn the alignment of a simple high-level model (a binary variable for toxicity present vs. toxicity not present) with the low-level neural representation (an individual layer of the LLM), and then perform the intervention in the projected space at inference time, once a good alignment has been learned through training the projection matrix?

---

> ### Author Response · Authors · 2024-11-29
>
> Dear Reviewer CvQ7,
>
> Thank you again for your positive review and the helpful comments. During the discussion period we revised our related work section, updated paragraphs following the reviewers' suggestions, explained the choice of linear maps over more complicated ones, included details on computational complexity and discussed data selection techniques.
>
> We would like to draw your attention to additional sets of new experiments conducted. Reviewer **WjWp** suggested showing the preservation of distributions empirically as well as running an ablation study on the pooling operation; both experiments have been included in the manuscript. Reviewer **eCSW** suggested evaluating diversity, thus we computed the Self-BLEU scores on generated sentences, showing that Linear-AcT offers better diversity than ITI.
>
> We remain available until the end of the rebuttal period to answer any new question or concern you may have.

---

### Official Review · Reviewer_Rxai · 2024-11-06

**Soundness:** 4
**Presentation:** 4
**Contribution:** 4
**Rating:** 8
**Confidence:** 3

**Summary:**

The paper introduces Activation Transport (ACT), a framework based on optimal transport theory to steer model activations and control generative model outputs. ACT is designed to modify activations from a source distribution (e.g., toxic language) to a target distribution (e.g., non-toxic language) while preserving the natural activation distributions within the model. The framework applies to both large language models (LLMs) and text-to-image diffusion models (T2Is), supporting tasks like toxicity mitigation, concept induction, style control, and concept negation. ACT outperforms other methods in robustness and interpretability by allowing fine-grained, interpretable control of generative behaviors with minimal computational overhead.

**Strengths:**

1. ACT is a simple and efficient transport function approach that seems to perform well on the experimental setups for both LLM and T2I without significant impact on performance.
2. The paper is well written with clear and easy to follow formulation and experimental results, The paper demonstrates ACT’s effectiveness in diverse tasks, including toxicity mitigation, concept induction, style control, and concept negation, showing superior or comparable performance to existing methods. The method’s flexibility and consistent results across both LLM and T2I applications underscore its potential as a general-purpose activation steering tool.
3. By basing the intervention on optimal transport theory, ACT provides a clear, interpretable parameter (λ) that adjusts the strength of control. This parameterisation allows for easy tuning and understanding of model adjustments, enhancing its usability for practitioners.

**Weaknesses:**

1. ACT currently relies on linear transport maps, which are computationally efficient but may not capture complex, non-linear relationships within activations, especially in large or multimodal generative models. This assumption could limit its effectiveness in applications requiring nuanced adjustments.

2. The quality of ACT’s transport maps depends on the representativeness of the source and target samples. If the samples do not fully capture the intended distribution (e.g., all aspects of toxic vs. non-toxic language), the intervention may be less accurate, impacting model behavior under real-world conditions with unseen data.

3.  Although ACT shows promising results in mitigating toxicity and inducing specific concepts, the paper provides less evidence of its effectiveness across a broader range of behavioral modifications, especially in challenging or ethically sensitive areas like misinformation suppression or bias control.

4. The performance of ACT can be influenced by which model layers are selected for intervention. While the paper provides some guidance, a more systematic approach to identifying optimal layers would improve robustness and reduce the need for manual tuning.

**Questions:**

1. A potential future work could be the focus on data selection to estimate the linear transport. This I would argue might produce better improvements than moving to non-linear estimations

2. although you show good evidence of the relatively minor degradation of model performance, more evidence is needed in a more quantifiable way, for example by showing limited impact on downstream tasks

---

> ### Author Response · Authors · 2024-11-20
> **Rebuttal to Reviewer Rxai [1/3]**
>
> We would like to thank you for your time, your encouragements and the many comments and questions provided. This has helped us improve our manuscript, through the answers below and additional experiments.
>
> > **W1. ACT currently relies on linear transport maps, which are computationally efficient but may not capture complex, non-linear relationships within activations, especially in large or multimodal generative models. This assumption could limit its effectiveness in applications requiring nuanced adjustments.**
>
> Thanks for raising this point. Our maps are linear (in fact affine), but more specifically, they are coordinate-wise. They can be cast as $T(x) = \omega \circ x + b$, element-wise multiplication + translation. Within the space of all OT map estimators studied in the literature so far, this is indeed a strong restriction.
>
> However, we have ran into four important problems when trying to add complexity to these estimators. We enumerate them from _easiest_ to fix (potentially) to _most challenging_:
>
> - **Computational**. Linear-ACT is easy to compute (Def. 3.1, L. 190) and only requires sort and mean operations. Given that we estimate repeatedly OT maps, at multiple layers, more advanced map estimators (e.g. Sinkhorn map estimator [1] or ICNNs [2]) would consume far more compute.
>
> - **Robustness**: We operate at high dimensional regimes (e.g. $d= 4096$), where $d$ is sometimes of the order of magnitude of samples. This is known to be a challenging regime for OT map estimation (the curse of dimensionality mentioned now in L.141). By contrast, estimating OT in 1D (as we do repeatedly across neurons) is very stable and has favorable statistical rates in $1/n$.
>
> - **Inference time/storage  overhead**: Our intervention is intended to be used at inference time. With a fixed $\lambda$ activation strength, Linear-ACT incurs no overhead, since the modifications can be merged in MLP weights. When the user wants to adjust strength $\lambda$, Linear-AcT requires storing 2 parameters $\omega,\beta$ per neuron. Any OT map estimator with more parameters would significantly inflate the execution/memory footprint for inference (regardless of $\lambda$).
>
> - **Compositionality**:  In our causal approach, we use previously estimated OT maps in lower level layers to impact the transport estimation for the next layer (Section 3.2). This compositionally requires some form of stability of OT maps. In effect, any poorly estimated OT map at a lower layer ruins the estimation for all subsequent layers. In that context, Linear-ACT has medium bias (it does not reconstruct perfectly the train target from train source) but low variance (it won’t fail spectacularly out-of-sample). In higher dimensions, because of the curse of dimensionality, most OT estimators won't generalize well, and will have high-variance. While this might be ok for a single layer (as most paper consider), it is going to be a lot more problematic when composition is involved.
>
> We spent quite some time investigating more advanced approaches, e.g. with cross-variable effects, but we discarded them as they yielded worse results and required more hyper parameter tuning.
>
> Taken together, this is why we settled on something simple, robust, and easy to interpret. That said, we agree with the reviewer that more sophisticated maps that can overcome the challenges above provides a very interesting research opportunity
>
> We have added a short discussion on this (L.142 in revision, highlighted in blue) and will expand it the appendix.
>
> [1] Entropic estimation of optimal transport maps. Aram-Alexandre Pooladian, Jonathan Niles-Weed, arXiv:2109.12004
>
> [2] On amortizing convex conjugates for optimal transport. Brandon Amos. arXiv:2210.12153
>
> > **W2. The quality of ACT’s transport maps depends on the representativeness of the source and target samples. If the samples do not fully capture the intended distribution (e.g., all aspects of toxic vs. non-toxic language), the intervention may be less accurate, impacting model behavior under real-world conditions with unseen data.**
>
> Thanks for raising this excellent point. We agree with you, data quality is of key importance. We believe this question is closely tied to the previous one (robustness of OT map estimation in the presence of noise in samples). For now, we posit that variability in data should be handled by robustifying or regularizing the map estimation pipeline. This is one argument in favour of independent 1D OT which won’t certainly be an optimal OT map, but which will do a reasonable job of pushing activations, *throughout layers*.
>
> One possible avenue to further robustify the estimation is to change the cost in L.185. We have used the squared-Euclidean distance on the real line, but many other costs $c$ would work, e.g. absolute deviation or $p$-norm. This would require running the 2D minimization in L. 187 per neuron, rather than getting closed forms as in L.187. We are exploring this choice.

---

> ### Author Response · Authors · 2024-11-20
> **Rebuttal to Reviewer Rxai [2/3]**
>
> > **W3. Although ACT shows promising results in mitigating toxicity and inducing specific concepts, the paper provides less evidence of its effectiveness across a broader range of behavioral modifications, especially in challenging or ethically sensitive areas like misinformation suppression or bias control.**
>
> Thanks for this great comment.
>
> In relation to ethically and sensitive areas, we have demonstrated that ACT is generic enough to be used as a **blueprint** by providing results on reducing toxicity in LLMs (Sec 4.1), inducing truthfulness in LLMs (Sec 4.3), and for diffusion models we have shown the ability to prevent specific concepts from being generated in images (Sec 5.2).
>
> To avoid using images that could hurt the sensitivity of the reader we used playful concepts such as “pink elephant”. However, a straightforward application for safety would be, for example, preventing generation of images that contain nudity or any other “not suitable for work” content.
>
> We agree that there are potentially many more applications that could benefit from this intervention that we did not explore. However, thanks to ACT's reliable and robust pipeline, we believe they will soon be tested in future works and applications.
>
> > **W4. The performance of ACT can be influenced by which model layers are selected for intervention. While the paper provides some guidance, a more systematic approach to identifying optimal layers would improve robustness and reduce the need for manual tuning.**
>
> On **Table 8** we report an ablation on Gemma-2-2b by layer type, showing that best results are obtained when intervening upon LayerNorm layers. In general, we found LayerNorm layers to achieve consistent results across models and tasks. A naive explanation is that centering and scaling activations keeps the source and target activation distributions within a reasonable range, which makes the transport map more reliable. For Stable Diffusion, choosing the LayerNorm was natural, since the denoising UNet is already conditioned through the LayerNorm layers, which is common in image generation [1, 2].
>
> To futher clarify the layer choice guidance, we have added a comment in **L.370**. We will expand this discussion further in the appendix.
>
> A related ablation to layer choice is the pooling choice, which has been suggested by reviewer **WjWp**. To complement the experiments in the paper, we run new experiments (Appendix D in the revision) explicitly showing that mean-pooling offers the best trade-off between conditioning strength and model utility preservation.
>
> > **Q1. A potential future work could be the focus on data selection to estimate the linear transport. This I would argue might produce better improvements than moving to non-linear estimations**
>
> This is a great point. We believe this is also tied to robustness, and might yield an interesting tradeoff where one wants to capture dominant changes while not being impacted by outliers. Beyond the choice of a different cost that you have raised above, we are currently considering running *unbalanced* transport map estimators, which can downweight automatically datapoints that are harder to transport (see early reference [3] in the context of NN based _multivariate_ OT map estimation).
>
> Fortunately, in our case, keeping the assumption that we use maps that decompose along dimensions, this would translate into unbalanced 1D map estimations (1 for each Neuron) to get more robust estimates for $\omega, \beta$. Practically, instead of a simple sum of squares between sorted values as in L.189, we would need to run [Alg. 2 in 3] to compute this unbalanced loss and differentiate its optimal value (using Danskin/Enveloppe theorem, therefore no overhead) w.r.t. $\omega, \beta$ to run gradient descent.
>
> [3] Scalable Unbalanced Optimal Transport using Generative Adversarial Networks, Karren D. Yang, Caroline Uhler, arXiv:1810.11447
>
> [4] Faster Unbalanced Optimal Transport: Translation invariant Sinkhorn and 1-D Frank-Wolfe. Thibault Séjourné, François-Xavier Vialard, Gabriel Peyré, https://arxiv.org/abs/2201.00730

---

> > ### Author Response · Authors · 2024-11-20
> > **Rebuttal to Reviewer Rxai [3/3]**
> >
> > > **Q2. Although you show good evidence of the relatively minor degradation of model performance, more evidence is needed in a more quantifiable way, for example by showing limited impact on downstream tasks**
> >
> > Showing evidence of limited impact on downstream tasks is indeed of paramount importance for any intervention method. This is why we evaluated the LLMs before and after our intervention on the MMLU benchmark. MMLU includes 57 downstream tasks including STEM, the humanities, the social sciences, laws, and more. It has a range of difficulties (from an elementary level to an advanced professional level), and it tests both world knowledge and problem solving ability.
> >
> > Linear-Act has an impact on MMLU that  ranges between 0.5 to 1.8 (depending on the model and the type of intervention, refer to Table 2 and 3). Thank you for highlighting the importance of downstream tasks, in the paper we will clarify what the MMLU benchmark is and why it is a valuable test for downstream tasks. We are keen to other about other similar benchmarks, yet complementary to MMLU, that you think are worth considering.
> >
> > For diffusion models the downstream tasks are less well defined, in the absence of a solid benchmark we have evaluated the impact of the intervention on the zero-shot-classification task. Similarly, we are potentially very interested in any other suggestion in the diffusion space.

---

### Author Response · Authors · 2024-11-25
**General comment**

We would like to thank the Reviewers and the Area Chairs for their time and effort assessing our submission. We are grateful for the many encouragements we have received:

-   *“ACT is a simple and efficient transport function approach”, “The paper is well written with clear and easy to follow formulation and experimental results”, “The paper demonstrates ACT’s effectiveness in diverse tasks”, “The method’s flexibility and consistent results [...] underscore its potential as a general-purpose activation steering tool.”*  **(Rxai)**
-   *“The experiments are extensive, covering a wide range of control tasks [...]”, “Many of them are important tasks”* **(CvQ7)**
-   *“The proposed approach is grounded, intuitive and is applicable to several modalities”“paper is well written and easy to follow”. “Extensive experiments”* **(WjWp)**
-   *“The use of optimal transport theory to steer activations is a novel idea that generalizes previous activation-steering methods.”*  **(eCSW)**

The reviewers have therefore highlighted **the broad applicability of our method (Rxai, WjWp. CvQ7, eCSW)**, but also the fact that our method **generalizes existing methods (CvQ7, eCSW)** with a paper that is **well-written and easy to follow (Rxai, WjWp)**.

**Paper Updates**
We would like to thank reviewers again for their constructive criticism as it has certainly helped us to improve the submission. We have taken into consideration each of the reviewers’ comments and here is a summary of the changes/additions we have made to address them:


-   **Impact of transport on diversity of generated samples**: We have introduced a diversity evaluation based on the self-BLEU score which shows that linear-AcT excels at preserving the original model diversity compared to other interventions. We would like to thank **eCSW** for asking this insightful question. Results can be found in Appendix G.
-   **Pooling operation ablation**: We have run an ablation experiment on the choice of token pooling operation that supports our choice of mean pooling (Section 3.3), and we have placed it in Appendix D.
-   **Insights on how linear-Act preserves activation distributions:** We have included plots of activation distributions before and after applying the transport map in Appendix F. We found that linear-AcT at $\lambda=1$ tends to transport activations within the target distribution, which is not the case for other methods.
-   **Detailed computational complexities**: We have extended Appendix A with detailed computational complexities for (i) obtaining activations, (ii) estimating transport maps, and (iii) applying transport maps at generation time. We highlight that linear transport maps can be fused into model weights, in which case there are no additional costs when generating with linear-AcT.
-   **Discussion about linear and non-linear maps**: We have added a short discussion on this within the current space limits (L.142 in revision, highlighted in blue) and will expand it in the appendix considering the multiple challenges that using a non-linear map entails in terms of compute, robustness, memory requirements, and compositionally of the interventions.
-   **Additional clarifications:** we would like to thank reviewers for giving us the opportunity to clarify some of the questions they had while assessing our work. We have replied to each of the reviewer’s questions and clarified each of the points in the submission (highlighted in blue).

We remain available until the end of the rebuttal period to answer any new question or concern you may have. Thanks again for the time you have spent reading our paper and our rebuttal.

---

### Meta-Review · Area_Chair_gPGk · 2024-12-20

**Metareview:**

The paper addresses the challenge of activation steering for controlling generative models by learning an optimal transport map between source and target activation distributions while preserving the target distributions. The authors highlight that existing activation-steering approaches are equivalent to mean transport maps and lack robustness across different models, layers, and tasks. To address these limitations, the authors propose Linear-AcT, which performs linear transportation between i.i.d. activations and addresses the issue of inducing OOD activations by introducing the transport support. This approach enables effective inference-time intervention with negligible computational overhead and allows for straightforward control of the intervention degree using a simple parameter, 𝜆. The experimental results demonstrate the effectiveness of Linear-AcT and its modality-agnostic nature through extensive evaluations on large language models and text-to-image diffusion models. Despite its strengths, the paper has certain weaknesses. By simplifying the mapping to a linear approach, it potentially overlooks the potential benefits of non-linear mappings, which could capture more complex relationships and allow for finer control. Furthermore, while the introduction of transport support to mitigate OOD activations is presented as a key contribution, the experimental evidence supporting this claim is insufficiently detailed. However, during the rebuttal phase, the authors addressed these concerns convincingly by providing additional experimental evidence and clarifying their approach. The combination of methodological novelty, practical effectiveness, and experimental results across diverse tasks solidifies the contribution of this work and leads to the basis for its acceptance.

**Additional Comments On Reviewer Discussion:**

All reviewers agreed that the proposed Linear-AcT is an efficient approach for controlling generative models, with extensive experiments on large language models and text-to-image diffusion models providing sufficient evidence for its effectiveness. However, most reviewers highlighted the lack of discussion regarding non-linear maps. The authors addressed this concern by explaining that they had already investigated non-linear maps but found them to perform worse and be less effective due to the increased number of hyperparameters. Recognizing the importance of this feedback, the authors added a discussion on this to the paper. Another significant point raised was the need for further experimental validation to support claims. For example, reviewer WjWp commented that the experiments to the claim of AcT better preventing representations from being OOD is lacking and Reviewer eCSW commented that the detailed analysis of computational efficiency is lacking although the proposed method claims to have low computational overhead. The authors conducted additional experiments and provided results that addressed these points.

While three of the four reviewers did not respond to the rebuttal, I believe the authors’ rebuttal sufficiently addressed the raised question. Consequently, it is reasonable to expect that three unresponsive reviewers would maintain their scores and one responding reviewer increased the score from 6 to 8, indicating a positive reception of the authors' responses. Given the reviewers’ recommendations (8 - accept, 6 - weak accept, 8 - accept, 8 - accept), the rebuttal's effectiveness in resolving raised concerns, and the overall strength of the paper, I recommend accepting this submission.

---

### Decision · Program_Chairs · 2025-01-22

Accept (Spotlight)